# Vec2Face: Scaling Face Dataset Generation with Loosely Constrained Vectors

**Haiyu Wu[1], Jaskirat Singh[2], Sicong Tian[3], Liang Zheng[2], Kevin W. Bowyer[1]**
[1]University of Notre Dame, [2]The Australian National University,[3]Indiana University South Bend
`{hwu6@nd.edu, jaskirat.singh@anu.edu, tiansic@iu.edu}`
`{liang.zheng@anu.edu, kwb@nd.edu}`

## Abstract

This paper studies how to synthesize face images of non-existent persons, to create a dataset that allows effective training of face recognition (FR) models. Besides generating realistic face images, two other important goals are: 1) the ability to generate a large number of distinct identities (inter-class separation), and 2) a proper variation in appearance of the images for each identity (intra-class variation). However, existing works 1) are typically limited in how many well-separated identities can be generated and 2) either neglect or use an external model for attribute augmentation. We propose Vec2Face, a holistic model that uses only a sampled vector as input and can flexibly generate and control the identity of face images and their attributes. Composed of a feature masked autoencoder and an image decoder, Vec2Face is supervised by face image reconstruction and can be conveniently used in inference. Using vectors with low similarity among themselves as inputs, Vec2Face generates well-separated identities. Randomly perturbing an input identity vector within a small range allows Vec2Face to generate faces of the same identity with proper variation in face attributes. It is also possible to generate images with designated attributes by adjusting vector values with a gradient descent method. Vec2Face has efficiently synthesized as many as 300K identities, whereas 60K is the largest number of identities created in the previous works. As for performance, FR models trained with the generated HSFace datasets, from 10k to 300k identities, achieve state-of-the-art accuracy, from 92% to 93.52%, on five real-world test sets (*i.e.*, LFW, CFP-FP, AgeDB-30, CALFW, and CPLFW). For the first time, the FR model trained using our synthetic training set achieves higher accuracy than that trained using a same-scale training set of real face images on the CALFW, IJBB, and IJBC test sets.

## 1 Introduction

We aim to synthesize face images in a way that enables large-scale training sets for FR models, which have the potential to address privacy issues arising with web-scraped datasets of real face images (DeAndres-Tame et al., 2024; Melzi et al., 2024; Shahreza et al., 2024). It is generally recognized that a good training set for FR should have high inter-class separability (Kim et al., 2023; Boutros et al., 2023a) and proper intra-class variation (Qiu et al., 2021; Boutros et al., 2022a; Kim et al., 2023; Boutros et al., 2023a).

However, existing methods lack flexibility in controlling the generation process, leading to unsatisfactory inter-class and intra-class results. On the one hand, (Boutros et al., 2023a) points out that identities generated in (Qiu et al., 2021; Boutros et al., 2022b;a; 2024) have relatively low separability because identity generation is controlled by either a single coefficient (Deng et al., 2020) or by hard class labels. On the other hand, while it is useful to employ identity features as a condition to increase inter-class separability (Boutros et al., 2023a; Papantoniou et al., 2024), it is non-trivial to increase intra-class variation without the use of a separate model such as ControlNet (Zhang et al., 2023) or a model for style transfer.

In light of the above, this paper proposes Vec2Face, a holistic model enabling the generation of large-scale FR datasets. In inference, the workflow of Vec2Face is similar to (Papantoniou et al.,

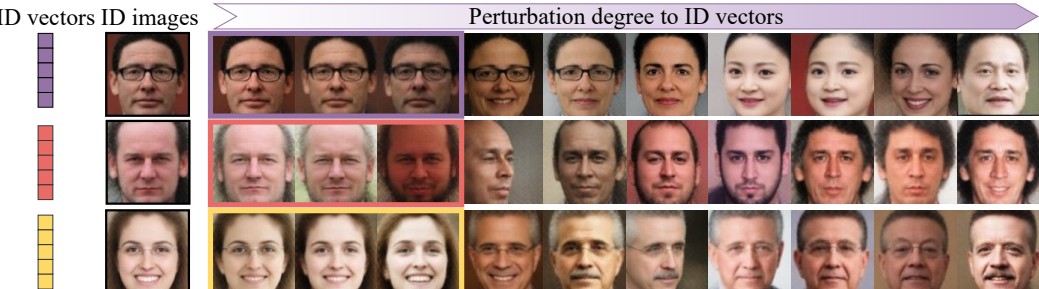

Figure 1: Example images generated by Vec2Face. From a random vector, we generate a face (ID image). We then perturb this vector with random values to generate diverse face images. Larger perturbation added to this vector results in larger dissimilarity to the ID images. The images in the frame are likely of the same people as the ID image.

2024; Boutros et al., 2023a), which uses a random vector as input and outputs a face image. But the *interesting* property of Vec2Face is that a small perturbation of the input vector leads to a face image of the same identity but with small changes in appearance, *e.g.*, pose, age, and facial hair and that a large perturbation of the input vector produces a face image of a different identity (Fig. 1). With this property, and because of the high-dimensionality (*i.e.*, 512-dim) of the input vectors, we can easily 1) sample a large number of well-separated identity vectors by controlling their similarity with each other to be under a certain threshold and 2) synthesize a large number of face images with various attributes by perturbing the identity vectors within a small range. Additionally, we can guide the image content at time of generation using a gradient descent method to efficiently generate faces with designated attributes, such as extreme pose angles and good image quality.

Using a proper cosine similarity threshold (*i.e.*, 0.3) to constrain the identity vector sampling, Vec2Face transfers the similarity between vectors to the similarity between generated images, resulting in large inter-class separability; intra-class variation is also properly large because we control the variance of the noise for perturbation, so the perturbed identity vectors have $\geq 0.5$ similarity with the identity vector, exhibiting various appearances while preserving the identity on the generated images. Experimentally, we verify these inter-class and intra-class variations are beneficial to the FR model performance (in Section 4.4). Because of this, FR models trained on the synthesized 10K identities and 500K images demonstrate very competitive accuracy compared with the state-of-the-art synthetic databases of the same scale (Kim et al., 2023; Papantoniou et al., 2024). Importantly, we show that scaling up the synthetic training data to 15 million images and 300K identities leads to further accuracy improvement. It is also worth noting that results on CALFW (Zheng et al., 2017), IJBB (Whitelam et al., 2017), and IJBC (Maze et al., 2018) represent the first instance where a model trained with synthetic data yields higher accuracy than that trained with real data at the same scale. To our knowledge, this is the first time a synthetic dataset can be this large and useful.

The training of Vec2Face aims to let a vector generate a face and let its perturbation magnitudes reflect face change magnitudes. To this end, in training we use vectors extracted from real face images by a pre-trained FR model. These vectors are fed into a feature masked autoencoder architecture, after which an image decoder produces images similar to the real ones at the pixel level. We summarize the main points of this paper below.

- We propose Vec2Face, a face synthesis model that allows us to efficiently synthesize a large number of face images and well-separated identities. Under an appropriate similarity threshold, Vec2Face effectively increases inter-class separability and intra-class variation of synthesized face images.

- With a generated dataset of 10K identities and 500K images, we achieve state-of-the-art FR accuracy measured as the average on five real-world test sets compared with existing synthetic datasets. Our performance even surpasses the accuracy obtained by real training data on three test sets. We also report that scaling to larger synthetic training data leads to further improvement.

## 2 RELATED WORK

**Discrete class label conditioned image generation.** VQGAN (Esser et al., 2021), MAGE (Li et al., 2023), DiT (Peebles & Xie, 2023), U-ViT (Bao et al., 2022), MDT (Gao et al., 2023), and VAR (Tian

et al., 2024) control the object classes of generated images using discrete class labels as condition. Therefore, these methods do not support the new class generation and are not applicable to the context of scalable face identity generation. Differently, Vec2Face uses continuous face features as input and thus technically can generate infinite new identities given appropriate feature sampling.

**Identity feature conditioned image generation.** Using a static feature as condition to guide the model to generate images for one identity is another approach. (Xiao et al., 2023; Chen et al., 2023; Li et al., 2024b) use CLIP (Radford et al., 2021) to encode the face embedding, but this representation is limited by CLIP's ability on facial encoding (Papantoniou et al., 2024). To improve the identity fidelity of the generated images, (Wang et al., 2024; Valevski et al., 2023; Ye et al., 2023; Wood et al., 2021; Papantoniou et al., 2024) use identity features extracted by a FR model as condition. However, an external model (*e.g.*, ControlNet (Zhang et al., 2023)) is needed to increase the attribute variation. Different from them, Vec2Face generates images for one identity using dynamic perturbed identity features that have high similarity value with their identity feature. This keeps a high identity fidelity while providing a way to control the attribute variation without using any external models.

**Synthetic face image datasets.** Existing approaches are primarily either GAN-based or diffusion-based. For the former, SynFace (Qiu et al., 2021), Usynthface (Boutros et al., 2022b), SFace (Boutros et al., 2022a), SFace2 (Boutros et al., 2024), and ExfaceGAN (Boutros et al., 2023b) leverage pre-trained GAN models to generate datasets. These methods are less effective because the pre-trained GAN models were not trained in an identity-aware manner. Departing from them, our GAN-based method explicitly uses face features in training, so the generated images are identity-aware.

For diffusion model-based methods, DCFace (Kim et al., 2023) uses a strong pre-trained diffusion model (Choi et al., 2021) and an additional style-transfer model to increase the identity separability and attribute variation. Again, this pre-trained model is not identity-aware. IDiff-Face (Boutros et al., 2023a) combines the pre-trained encoder and decoder from VQGAN (Esser et al., 2021) with a latent diffusion model conditioned by identity features to control the separability of the generated identities. Although the latent diffusion model is identity-aware, the pre-trained VQGAN compromises such effect. Arc2Face (Papantoniou et al., 2024) fine-tunes a pre-trained stable diffusion model (Rombach et al., 2022) on the WebFace42M dataset to increase the generalizability on face. It combines the FR feature and CLIP (Radford et al., 2021) feature to control the identity of output images. The pose variation is increased by adding an additional ControlNet (Zhang et al., 2023). However, neither the pre-trained stable diffusion model nor CLIP are optimized for face, and the slow processing speed strongly limits large scale dataset generation. Conversely, our method is specifically designed for face dataset generation and is efficient and identity-aware. This allows our method to easily scale the dataset size to 15 million images and achieve state-of-the-art FR accuracy.

## 3 METHOD

### 3.1 VEC2FACE: ARCHITECTURE AND LOSS FUNCTION

As shown in Fig. 2, Vec2Face consists of a pretrained FR model, a feature masked autoencoder (fMAE), an image decoder, and a patch-based discriminator (Isola et al., 2017; Yu et al., 2022).

**Feature extraction and expansion.** Given a real-world face image, following (Boutros et al., 2023a; Papantoniou et al., 2024), we compute its feature using a FR model. To match the input shape of $MAE_f$, the extracted image feature is projected and expanded to a 2-D feature map.

**Feature masked auto-encoder (fMAE).** Similar to MAE (He et al., 2022), the model is forced to learn better representations by masking out the input. Different from the MAE that masks an input image and introduces mask tokens to form the full-size image, the fMAE uses a 2D feature map as input and masks this feature map. To provide more useful information, the projected image feature is used to form the full-size feature map. Specifically, the rows in the feature map are randomly masked out by $x\%$ before the encoding process, where $x\% \in \mathcal{N}_{truncated}(max = 1, min = 0.5, mean = 0.75)$, and the projected image feature is filled in the masked out positions to form the full-size feature map before being processed by the decoder. The structure is in the Appendix.

**Image decoder and patch-based discriminator.** Finally, the new feature map is passed through a simple image decoder to generate/reconstruct the image. To improve image quality, a patch-based discriminator (Isola et al., 2017; Yu et al., 2022) is integrated to form a GAN-type training.

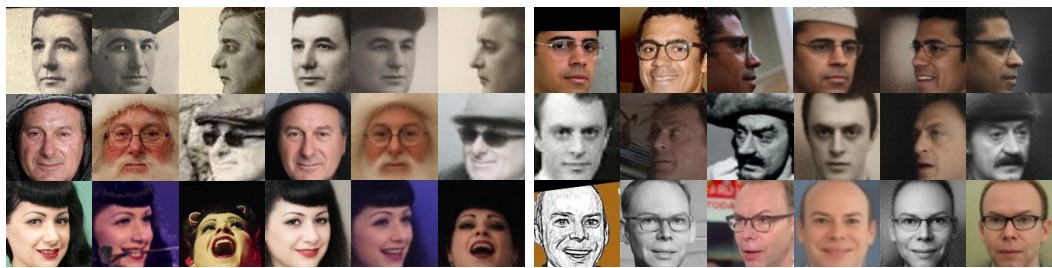

Figure 2: Architecture of Vec2Face. Given a real face image, we compute its feature "IM feature" using a face recognition model. This feature is projected and expanded into a feature map, and the latter is processed by a feature masked autoencoder (fMAE). Inside the fMAE, the rows in the feature map are randomly masked out before being processed by the encoder. The projected image feature is then used to form the full-size feature map before being processed by the decoder. Finally, the decoder outputs a feature map. Outside the fMAE, a small image decoder reconstructs the pixels in the original image based on the output of fMAE. During inference, Vec2Face accepts a randomized vector and generates a face image. This process has properties demonstrated in Fig. 1.

Original training          Reconstructed          Original unseen          Reconstructed

Figure 3: Reconstruction results for training (**left**) and unseen (**right**) identities. Specifically, we extract features of original images by using a FR model and feed them to Vec2Face for reconstruction. We observe that the reconstructed images still maintain the same identity while removing some image borders and backgrounds and transferring sketches into photo-realistic images.

**Loss function.** The training objective function includes an image reconstruction loss, an identity loss, a perceptual loss and a GAN loss. The reconstruction loss can be written as,

$$L_{rec} = MSE(IM_{rec}, IM_{gt}), \tag{1}$$

which compares the pixel-level difference between the reconstructed image $IM_{rec}$ and the ground truth images $IM_{gt}$. The identity loss is written as:

$$L_{id} = 1 - CosineSimlarity(FR(IM_{rec}), FR(IM_{gt})), \tag{2}$$

meaning that the features of the reconstructed image and original image extracted using the FR model should be close. In addition, the perceptual loss (Zhang et al., 2018) is used to ensure the correct face structure at the early training stage and a patch-based discriminator (Isola et al., 2017; Yu et al., 2022) is used to form the GAN loss to increase the sharpness of the generated images. The total loss is:

$$L_{total} = L_{rec} + L_{id} + L_{lpips} + L_{GAN}. \tag{3}$$

Some findings of the loss function's effects are: 1) without $L_{id}$, Vec2Face has slightly degraded generation performance but still works reasonably well on identity preservation; 2) without $L_{lpips}$, the reconstructed images do not have clear face structures; 3) using $L_{GAN}$ at early training stage makes the model convergence difficult, so it is used only after 1,000 epochs. Examples of loss effects on image reconstruction are in Appendix A.2.

**Image reconstruction results.** Fig. 3 shows that, for both seen and unseen identities, the generated images have very similar identities to the original images. Moreover, the reconstructed image would remove image border artifacts and some complex backgrounds and translate sketches into photo-realistic images.

### 3.2 INFERENCE: SAMPLING VECTORS TO GENERATE IDENTITIES AND THEIR FACE IMAGES

For Vec2Face, dissimilar input vectors will likely lead to images of different identities, allowing us to generate different identity images; controlling the perturbed vectors in a proper similarity range with their identity vector can generate face images with different attributes while preserving the identity. Thus, controlling the sampled vectors is important for dataset generation.

**Sampling well-separated identity vectors.** Suppose we want to generate $n$ identity images from $n$ identity vectors $v_1, v_2, ..., v_n$, where $v_i \in \mathbb{R}^d, i = 1, 2, ..., n$. We ask that the similarity between any pair of identity vectors be less than a threshold $\tau$: $sim(v_i, v_j) \leq \tau, \ \forall \ i \neq j$. This ensures that the generated images are of different identities. Following (Papantoniou et al., 2024), PCA is used to sample vectors. Specifically, we first compute the mean and covariance of the PCA-transformed face features, then use these statistics to sample vectors in the PCA space via multivariate normal distribution, and finally inverse transform these vectors back to the original feature space to obtain new identity vectors. Newly obtained identity vectors that have similarity $\leq \tau = 0.3$ with all existing vectors are kept, otherwise they are dropped. Interestingly, we notice that, in the high-dimensional (512 in this paper) feature space, most of the randomly sampled vectors fit the condition due to sparsity. In fact, only 1.7% of the sampled 300k vectors are filtered out.

**Perturbing identity vectors for image creation.** After obtaining $n$ identity vectors $v_1, v_2, ..., v_n$ , we sample $m$ vectors for each identity vector by perturbation:

$$v_{im_k} = v_n + \mathcal{N}(0, \sigma), \ \ k = 1, 2, ..., m, \tag{4}$$

where $v_{im_k}$ is the $k$th perturbed vector for an identity vector $v_n$. $\mathcal{N}(0, \sigma)$ is a Gaussian distribution with 0 mean and $\sigma$ variance. To ensure the identity-consistency after perturbation, the similarity values between perturbed vectors and their identity vectors are $\geq 0.5$.

Technically, the above steps allow us to synthesize an unlimited number of identities and their images using loosely constrained vectors via Vec2Face. Hyperparameter $\tau$ is selected such that identities are well-separated, *i.e.*, high inter-class variation, although it does not act as a strong filter in practice. The selection of hyperparameter $\sigma$ should enable the generated images for an identity to have large enough variance while staying on the same identity, *i.e.*, high intra-class variation.

## 3.3 INFERENCE: EXPLICIT FACE ATTRIBUTE CONTROL

By default, Vec2Face generates images without explicit attribute control. This might lead to some attributes being under-represented, *e.g.*, too few profile head poses. To address this, we introduce attribute operation, or AttrOP, to guide vector perturbation so that additional images of desired attributes can be generated. This work focuses on face pose and image quality.

Inspired by (Singh et al., 2023), AttrOP controls the attributes of the generated images by simply adjusting the values in the feature vectors via gradient descent, a process shown in Algorithm 1. Specifically, we first set a target image quality $Q$ and pose angle $P$ and prepare a pretrained pose evaluation model $M_{pose}$, a pretrained quality evaluation model $M_{quality}$, and a FR model $M_{FR}$. Then, given an identity vector $v_{id}$ and a perturbed identity vector $v_{im}$, we generate a face image from an adjusted vector $v'_{im}$ and compute the following loss functions:

$$
\begin{aligned}
\mathcal{L}_{attrop} &= \mathcal{L}_{id} + \mathcal{L}_{quality} + \mathcal{L}_{pose} \text{ where,} \\
\mathcal{L}_{id} &= 1 - CosSim(M_{FR}(IM), v_{id}), \\
\mathcal{L}_{quality} &= Q - M_{quality}(IM), \\
\mathcal{L}_{pose} &= abs(P - abs(M_{pose}(IM))),
\end{aligned}
\tag{5}
$$

Because both $M_{pose}$ and $M_{quality}$ are differentiable, gradient descent can be used to adjust $v'_{im}$ to minimize $\mathcal{L}_{attrop}$. We finally use the adjusted $v'_{im}$ to generate images that exhibit the desired pose and image quality. Sample images optimized by AttrOP are shown in Fig. 4, where profile poses and various image quality levels can be seen.

## 3.4 DISCUSSION

**Novelty statement.** Vec2Face is novel in three aspects. First, it is new to add small/large perturbations to input vectors to generate images of the same/different IDs. In comparison, existing works typically use fixed ID vectors without an explicit mechanism to separate different IDs. Second, as a minor contribution, our GAN architecture realizes the above function, where the fMAE component prevents overfitting during training, similar to VAE (Kingma & Welling, 2013). Third, Vec2Face can seamlessly support attribute control during inference, while existing methods have to rely on external models for this purpose.

**Algorithm 1:** AttrOP

---

1 **Function** `AttrOP` ($v_{id}$, $v_{im}$, $M_{gen}$, $T$)**:**

    **Input:** a) $v_{id}$: sampled ID vectors,

              b) $v_{im}$: initial perturbed ID vectors,

              c) $M_{gen}$: Vec2Face model,

              c) $T$: the number of iterations

    **Required:** target quality $Q$, target pose $P$

    **Output:** a) $v'_{im}$: adjusted perturbed ID vectors

2     Condition models: $M_{pose}$, $M_{quality}$, $M_{FR}$

3     Initialization $v'_{im} = v_{im}$

4     **for** $t = T - 1, T - 2, ..., 0$ **do**

5         $IM = M_{gen}(v'_{im})$

6         Calculate $\mathcal{L}_{attrop}$ in Eq. 5

7         $v'_{im} = v'_{im} - \lambda \nabla_{v'_{im}} \mathcal{L}_{attrop}$

8     return $v'_{im}$

---

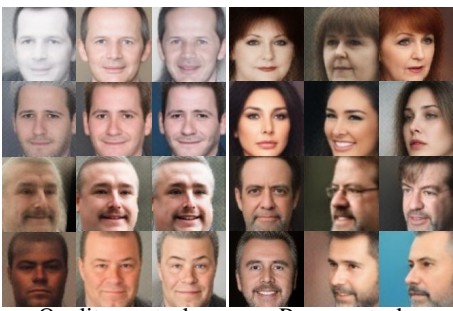

Quality control      Pose control

Figure 4: Sample generated images after image quality control (**left**) and head pose control (**right**). For each three-image group, from left to right, image quality increases, or head poses become more profile. This improves diversity of the synthetic dataset.

**Why can Vec2Face generate images like in Fig. 1?** The FR model maps images to a hyperspherical space, where features of the same identity are clustered, and those of different identities are farther apart (Deng et al., 2019). Using such features as input and supervised by the original images, Vec2Face learns how to generate face images based on characteristics of these vectors. As such, increasing the degree of the perturbation added to the vector gradually changes the vector direction, which results in faces of the same and then different identities.

**How can Vec2Face increase inter-class separability and intra-class variations?** A face feature vector reflects both identity and attributes. Hence, controlling the cosine similarity between sampled vectors can enforce separate identities, and adding small perturbations to sampled vectors could change the attributes while maintaining the identity. Moreover, AttrOP allows us to easily obtain useful vectors to generate images with desired attributes, further increasing intra-class variations.

**Can other structures work?** A diffusion model conditioned on image features may achieve similar function. However, the results in Appendix A.1 show that the initial noise image may exert a stronger influence on the final output than the conditioned image/identity features. This compromises identity-preserving capability. In comparison, the stochasticity of our method mainly comes from the image feature. fMAE has some randomness in testing, but has a very minor effect on generated images. That said, it would be interesting to design more effective architectures in the future.

**What is an identity?** An interesting philosophical question would be how to define an identity. Our research suggests that generated identities can be continuous, where a threshold $\tau$ is used to determine the boundary between different identities. We conducted a simple experiment for this in Section 4.4. It will be interesting to deeply study how this computational definition of identities connects with and differs from philosophical understanding in the future.

## 4 EXPERIMENTS

### 4.1 DATASETS AND EVALUATION PROTOCOL

**Vec2Face training set.** The training data consists of 1M images and their features from 50K randomly sampled identities in WebFace4M (Zhu et al., 2023), where the images features are extracted by an ArcFace-R100 model pretrained on Glint360K (An et al., 2021). Unless otherwise specified, this model is used for feature extraction throughout this paper.

**Real-world test sets.** There are eleven test sets used to compare the synthetic and real datasets on FR model training. LFW (Huang et al., 2008) tests the FR model in a general case. CFP-FP (Sengupta et al., 2016) and CPLFW (Zheng & Deng, 2018) test the FR model on pose variation. AgeDB (Moschoglou et al., 2017) and CALFW (Zheng et al., 2017) challenge the FR model with large age gap. These five test sets are used for a general accuracy comparison across synthetic datasets and a real dataset. Besides these five, Hadrian (Wu et al., 2024a) and Eclipse (Wu et al., 2024a) are used for intra-class variation evaluation, where Hadrian pairs emphasize facial hair difference

| Training sets | # ims | LFW | CFP-FP | CPLFW | AgeDB | CALFW | Avg. | IJBB | IJBC |
|---|---|---|---|---|---|---|---|---|---|
| IDiff-Face (Boutros et al., 2023a)[†] | 0.5M | 98.00 | 85.47 | 80.45 | 86.43 | 90.65 | 88.20 | 62.13 | 63.61 |
| DCFace (Kim et al., 2023)[†] | 0.5M | 98.55 | 85.33 | 82.62 | 89.70 | 91.60 | 89.56 | 66.47 | 69.92 |
| ID$^3$ (Li et al., 2024a)[†] | 0.5M | 97.68 | 86.84 | 82.77 | 91.00 | 90.73 | 89.80 | - | - |
| Arc2Face (Papantoniou et al., 2024)[†] | 0.5M | 98.81 | **91.87** | 85.16 | 90.18 | 92.63 | 91.73 | - | - |
| DigiFace (Bae et al., 2023)[★] | 1M | 95.40 | 87.40 | 78.87 | 76.97 | 78.62 | 83.45 | 5.42 | 6.10 |
| SynFace (Qiu et al., 2021)[◇] | 0.5M | 91.93 | 75.03 | 70.43 | 61.63 | 74.73 | 74.75 | - | - |
| SFace (Boutros et al., 2022a)[◇] | 0.6M | 91.87 | 73.86 | 73.20 | 71.68 | 77.93 | 77.71 | 37.90 | 41.94 |
| IDnet (Kolf et al., 2023)[◇] | 0.5M | 92.58 | 75.40 | 74.25 | 63.88 | 79.90 | 79.13 | - | - |
| ExFaceGAN (Boutros et al., 2023b)[◇] | 0.5M | 93.50 | 73.84 | 71.60 | 78.92 | 82.98 | 80.17 | - | - |
| SFace2 (Boutros et al., 2024)[◇] | 0.6M | 95.60 | 77.11 | 74.60 | 77.37 | 83.40 | 81.62 | - | - |
| **HSFace10K (Ours)**[◇] | 0.5M | **98.87** | 88.97 | **85.47** | **93.12** | **93.57** | **92.00** | **83.82** | **86.96** |
| CASIA-WebFace (Real) | 0.49M | 99.38 | 96.91 | 89.78 | 94.50 | 93.35 | 94.79 | 78.71 | 83.44 |

Table 1: Comparison of existing synthetic datasets on seven real-world test sets. †, ⋆, and ◇ represent diffusion, 3D rendering, and GAN approaches, respectively, for constructing these datasets. We also list the results of training on a real-world dataset CASIA-WebFace. Since previous works do not provide the IJB results, we re-train the FR models and report the accuracy on available datasets.

and Eclipse pairs differ on face exposure. The images in both datasets are indoor and high quality originating from MORPH (Ricanek & Tesafaye, 2006). In addition, SLLFW (Deng et al., 2017) and DoppelVer (Thom et al., 2023), are used to evaluate the identity definition used in the existing works (including ours). These two test sets have similar-looking pairs. Lastly, IJBB (Whitelam et al., 2017) and IJBC (Maze et al., 2018) are more challenging test sets that are closer to the real-world scenario.

**Evaluation protocol of test sets.** Besides IJBB and IJBC, the aforementioned test sets use the same evaluation protocol suggested in (Huang et al., 2008). Following the previous work (Deng et al., 2019), we use TPR@FPR=1e-4 to measure the model performance on IJBB and IJBC.

## 4.2 EXPERIMENT DETAILS

**Vec2Face model training:** We used the default ViT-Base as the backbone to form the fMAE. Since the image size of the commonly used FR training sets is 112x112 and the image decoder is a four-layer, double-sized architecture, the feature map size is set to 49x768, so that this feature map can be reshaped to 7x7x768 for image decoder to decode a 112x112x3 image. The optimizer is AdamW (Loshchilov & Hutter, 2017), with a learning rate of 4e-5 and a batch size of 32 per GPU. We use approximately 10 RTX6000 GPUs for each training run.

**Synthetic dataset generation.** We obtain well-separated identity vectors by PCA learned on face feature vectors of MS1MV2 (Deng et al., 2019), details are in Section 3.2. To get vectors for image generation, we perturb the identity vectors with the noise sampled from three Gaussian distributions. Specifically, 50 perturbed vectors are sampled for each identity, where 40% from $\mathcal{N}(0, 0.3)$, 40% from $\mathcal{N}(0, 0.5)$, and 20% from $\mathcal{N}(0, 0.7)$. But this resulting dataset suffers from the low pose variation, which causes bad performance on CFP-FP and CPLFW. To increase pose variation, we generate 30 images with large yaw angle via AttrOP for each identity and randomly replace the existing ones. In detail, the target pose $\mathcal{P}$ of 20 images is 60 and 10 images is 85. Meanwhile, we add the image quality control, $Q = 27$, to mitigate the quality degradation during pose adjustment.

**AttrOP details.** The MagFace-R100 (Meng et al., 2021), SixDRepNet (Hempel et al., 2024), and ArcFace-R100 are used as $M_{quality}$, $M_{pose}$, and $M_{FR}$ in Algorithm 1. $T$ is set to 5.

**Face synthesizing method comparison settings.** The standard training dataset size is 0.5M images from 10K identities. Unless otherwise specified, the backbone is SE-IR50 (He et al., 2016), the recognition loss is ArcFace (Deng et al., 2019), and other configuration details are in the Appendix.

## 4.3 MAIN EVALUATION

**Comparison with state-of-the-art synthetic datasets at the scale of 0.5M-0.6M images.** In Table 1, we compare FR accuracy of models trained with datasets synthesized by different methods, *i.e.*, diffusion models, GANs, and 3D rendering. We have the following observations. *First*, HSFace10K synthesized by Vec2Face yields very competitive accuracy: **98.87%, 88.97%, 85.47%, 93.12%, 93.57% on LFW, CFP-FP, CPLFW, AgeDB, and CALFW, respectively, and 92.00% on average**. Our method is only lower than Arc2Face on the CFP-FP dataset (88.97% vs. 91.87%) and is the

| Datasets | # images | LFW | CFP-FP | CPLFW | AgeDB | CALFW | Avg. |
|---|---|---|---|---|---|---|---|
| HSFace10K | 0.5M | 98.87 | 88.97 | 85.47 | 93.12 | 93.57 | 92.00 |
| HSFace20K | 1M | 98.87 | 89.87 | 86.13 | 93.85 | 93.65 | 92.47 |
| HSFace100K | 5M | 99.25 | 90.36 | 86.75 | 94.38 | 94.12 | 92.97 |
| HSFace200K | 10M | 99.23 | 90.81 | 87.30 | 94.22 | 94.52 | 93.22 |
| HSFace300K | 15M | 99.30 | 91.54 | 87.70 | 94.45 | 94.58 | **93.52** |
| HSFace400K | 20M | 99.37 | 90.53 | 87.35 | 94.33 | 94.60 | 93.24 |
| CASIA-WebFace (Real) | 0.49M | 99.38 | 96.91 | 89.78 | 94.50 | 93.35 | 94.79 |
| CASIA-WebFace + HSFace10K | 0.99M | **99.58** | **97.06** | **90.58** | **95.62** | **94.67** | **95.50** |

Table 2: Impact of scaling the proposed HSFace dataset to 1M images (20K IDs), 5M images (100K IDs), 10M images (200K IDs), 15M images (300K IDs), 20M images (400K IDs). Continued improvement is observed until 300K IDs. We also list the performance obtained by training on the real-world dataset CASIA-WebFace and its combination with HSFace10K. The latter combination yields even higher accuracy.

state of the art on all the other datasets and average. *Second*, on the CALFW, IJBB, and IJBC datasets, the accuracy of HSFace10K is higher than that of CASIA-WebFace (Yi et al., 2014). To our knowledge, this is the first time that a model trained with synthetic data outperforms that trained with same-scale real data. *Third*, generally speaking, GAN-based methods, while being prevalent, are not as competitive as diffusion-based and 3D rendering methods. Our method is GAN-based but outperforms the other methods.

**Effectiveness of scaling up the proposed HSFace dataset.** Vec2Face can easily generate a large-scale dataset by sampling more identity vectors which have a cosine similarity $\leq \tau$ with any other identity vectors. To test the efficacy of Vec2Face on scaling, we generate six datasets with increased identity numbers from 10K to 400K, where each identity has 50 images. Results of models trained on these datasets are shown in Table 2.

It is clear that scaling up our synthetic dataset leads to consistent accuracy improvements until 300K IDs: from 92.00%, to 92.47%, 92.97%, 93.22%, 93.52%. Notably, our HSFace300K is 12.5 times larger than the largest synthetic dataset prior to this paper while still giving steady accuracy improvement. These results clearly demonstrate the advantage of Vec2Face in data scaling. Also note that Vec2Face is only trained with 1M real-world images. We speculate that a Vec2Face model trained with larger initial real data could bring further improvements.

**Merging synthetic and real-world training sets.** In Table 2, after merging HSFace10K and CASIA-WebFace, we observe improvement over using either dataset alone for training. We obtain 95.50% average accuracy, which is higher than 92.00% (HSFace10K alone) and 94.79% (CASIA-WebFace alone). This indicates that HSFace10K has good quality and is complementary to real data. Additional experiments are in the Appendix.

**Effectiveness of attribute control.** Images generated by Vec2Face are mostly near-frontal, which can be attributed to such images being most frequent in the training set. Using AttrOP, we synthesize faces with some profile poses where we use target Yaw angles 60° and 85°. Results are shown in Table 3. We observe that adding faces with 60° poses leads to 3.46% improvement, while adding

| Attr. control | LFW | CFP-FP | CPLFW | AgeDB | CALFW | Avg. |
|---|---|---|---|---|---|---|
| - | 98.27 | 76.56 | 81.70 | 90.75 | 92.92 | 88.04 |
| + Quality 27 | 98.55 | 83.27 | 83.68 | 91.12 | 93.27 | 89.98 |
| + Angle 60° | 98.62 | 86.46 | **85.75** | 92.85 | **93.80** | 91.50 |
| + Angle 85° | **98.87** | **88.97** | 85.47 | **93.12** | 93.57 | **92.00** |

Table 3: Impact of face quality and pose control via AttrOP in FR accuracy. Each of the training sets has 0.5M images. The accuracy improvement is observed when more controls are added for image quality and pose angles, especially on pose-oriented test sets, CFP-FP and CPLFW.

85° gives further 0.5% improvement. These results demonstrate the effectiveness of attribute control. Unlike prior works (Kim et al., 2023; Papantoniou et al., 2024), this is achieved by a single model.

**Image generation cost and quality of reconstruction.** The resource requirement for image generation is an important metric, especially in this large model era. We compare our method with a state-of-the-art face generation model. Experimental details: 1) both models are tested on a single Titan-Xp, 2) the batch size is 8, 3) a 4-step scheduler (Luo et al., 2023) is

| Methods | Computing cost | | FID | |
|---|---|---|---|---|
| | Model size | FPS | LFW | Hadrian |
| Arc2Face | 3.4GB | 1.5 | 43.80 | 53.27 |
| Vec2Face | **0.68GB** | 467 | 35.75 | 51.93 |

Table 4: Computing cost and FID measurement of Arc2Face and Vec2Face.

used for Arc2Face. Table 4 shows that our method is 311x faster than Arc2Face. We also use FID to measure the distance between the original LFW and Hadrian images, and the reconstructed images from Arc2Face and Vec2Face, where the images are measured at 112x112 resolution. Our model slightly outperforms the Arc2Face results, in size of model, speed of generation, and fidelity to original images , as shown in Table 4. More results are in Appendix A.1.

## 4.4 FURTHER ANALYSIS

**Vec2Face generates datasets with large inter-class separability.** Since only Arc2Face and Vec2Face allow us to generate a large number of identities, we directly use the datasets (including a real dataset) released in previous works to calculate the identity vectors for separability evaluation. For Arc2Face and Vec2Face, we sample 200K well-separated vectors to generate identity images, where we report the results of using LCM-lora (Luo et al., 2023) for Arc2Face. Fig. 5 presents the number of identities whose cosine similarity against any other identity vector is less than 0.4, as more generated identities are gradually added for each dataset. Note that, a reasonable threshold for separability evaluation varies when using different FR models (Kim et al., 2023; Boutros et al., 2024). We select 0.4 is because it gives us a reasonable number (183K out of 200K) on a real dataset. From the results, we have three observations.

First, synthetic datasets such as SynFace and SFace generally have lower inter-class separability than real-world datasets. This is because their data synthesis methods do not utilize the face feature characteristics for image generation, as Arc2Face and IDiff-Face do. Second, Arc2Face has much larger identity separability than existing methods, comparing with the number reported in (Kim et al., 2023), but is still inferior to Vec2Face. This is because the initial noise image sometimes exerts a stronger influence on the final output than the conditioned identity vector. Third, real-world datasets have slightly lower identity separability than our method, because they have a small number of twins, close relatives, or even the same person with different character names in TV shows (Wu et al., 2024b). **Note that 0.4 is not used to evaluate the number of actual identities in the dataset, but rather to provide a consistent measurement of identity separability across datasets.** Overall, the special design of Vec2Face help us to generate well-separated identities.

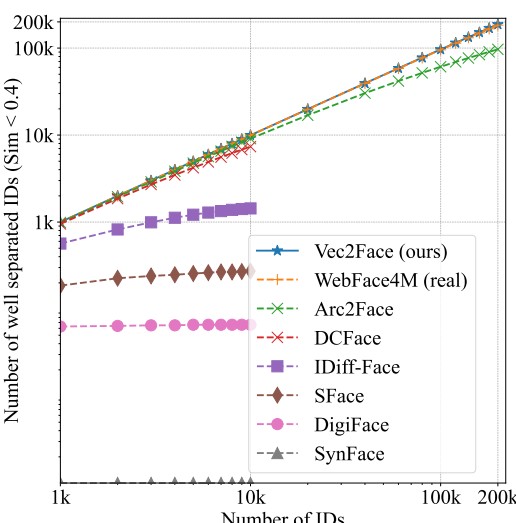

Figure 5: Comparing existing synthetic FR dataset generation methods on inter-class separability.

**Impact of inter-class separability on FR accuracy.** Table 7 summarizes how different average identity similarity or different levels of identity separability affect FR accuracy. Results indicate that high identity separability is beneficial, which is consistent with previous findings (Boutros et al., 2023a). A unique finding is that too large separation does not further benefit the performance.

**Vec2Face generates datasets with large intra-class variation.** The accuracy values in Table 1 indicate the generated dataset has large variation on age and pose. To evaluate the intra-class variation on other attributes, we test the models on Hadrian and Eclipse (Wu et al., 2024a), which have 6K image pairs for each and respectively emphasize variation on facial hair and face exposure. Table 6 shows that the datasets generated by Vec2Face have proper variation on these two attributes, as increasing the dataset size eventually surpasses the accuracy of the real dataset.

**Impact of hyperparameter $\sigma$.** In Vec2Face, $\sigma$ directly controls intra-class variation, *i.e.*, larger $\sigma$ means higher intra-class variation and vice versa. We test different configurations of $\sigma$ values and its sampling %. Table 5 shows that, when increasing $\sigma$, FR accuracy first increases and then drops. Initially, increasing intra-class variation is beneficial because it creates some hard training samples. But if $\sigma$ is too large, the generated face may look like a different person, degrading the accuracy. Similarly, increasing the sampling % of the large $\sigma$ may hurt the identity consistency in a folder,

| $\sigma$ | Sampling % | $Sim_{min}$ | Avg. |
|---|---|---|---|
| {0.3} | {100} | 0.71 | 82.71 |
| {0.3, 0.5} | {60, 40} | 0.62 | 86.25 |
| {0.3, 0.5, 0.7} | {40, 40, 20} | 0.53 | **88.04** |
| {0.3, 0.5, 0.7} | {20, 40, 40} | 0.51 | 87.26 |
| {0.3, 0.5, 0.9} | {40, 40, 20} | 0.47 | 85.75 |

Table 5: Impact of $\sigma$ values with corresponding sampling % for each identity on average FR accuracy (%). We use 0.5M images from 10K identities for training. AttrOP is not applied and $Sim_{min}$ is the minimum cosine similarity between the perturbed vectors and their ID vectors.

| Datasets | Hadrian | Eclipse | SLLFW | DoppelVer |
|---|---|---|---|---|
| HSFace10K | 69.47 | 64.55 | 92.87 | 86.91 |
| HSFace20K | 75.22 | 67.55 | 94.37 | 88.90 |
| HSFace100K | **80.00** | **70.35** | 95.58 | 90.39 |
| HSFace200K | **79.85** | **71.12** | 95.70 | 89.86 |
| HSFace300K | **81.55** | **71.35** | 95.95 | 90.49 |
| CASIA-WebFace | 77.82 | 68.52 | **96.95** | **95.11** |

Table 6: Comparing a real dataset with HSFaces on other tasks. Hadrian, Eclipse, SLLFW, and DoppelVer emphasize facial hair variation, face exposure difference, similar-looking, and doppelganger, respectively.

| $\tau$ | Avg. ID sim. | Avg. |
|---|---|---|
| | 0.88 | 64.62 |
| 0.3 | 0.70 | 77.65 |
| | 0.56 | 78.67 |
| | 0.01 | **88.04** |
| 0.2 | 0.003 | 86.60 |

| # IDs × # IMs | Avg. |
|---|---|
| 10K×5 | 75.02 |
| 10K×10 | 83.09 |
| 10K×20 | 89.45 |
| 10K×50 | 92.00 |
| 10K×80 | 92.03 |

| Architectures | Avg. |
|---|---|
| SE-IResNet18 | 89.60 |
| SE-IResNet50 | 92.00 |
| SE-IResNet100 | 92.49 |
| SE-IResNet200 | 92.49 |
| ViT-S | 88.54 |
| ViT-B | 90.15 |

Table 7: Impact of inter-class separability. Larger average identity similarity, or low separability reduces the accuracy. We use 0.5M images from 10K identities for training.

Table 8: Impact of number of images per identity. Different number of images are generated by the best $\sigma$ and their sampling % setting. The average accuracy (%) on five test sets is reported.

Table 9: HSFace10K training on ResNets and Vision Transformers (ViT). The average accuracy (%) on five test sets is reported.

which also degrades the accuracy. Thus, we choose $\sigma = \{0.3, 0.5, 0.7\}$ to sample $\{40\%, 40\%, 20\%\}$ images for each identity.

**Does Vec2Face generate new identities compared with its training identities?** We compute the feature similarity between the 300K identities from the proposed synthetic data and the 50K training identities selected from WebFace4M. We find that only 0.398%, or 1194 out of the 300K synthetic identities are similar to the training identities, with a similarity score greater than 0.4. To completely avoid using real identities for training FR models, we have replaced these 1194 identities with new identities, and the performance change is negligible. More analysis is in the Appendix.

**More images per identity is beneficial in FR accuracy.** In Table 8, increasing the number of images per identity from 5 to 80 increases the accuracy, but this improvement is saturated at 80.

**Larger backbone helps the FR model performance.** In Table 9, accuracy of six models trained with HSFace10K and ArcFace loss increases from SE-IResNet18 to SE-IResNet100 and is saturated at SE-IResNet200. Accuracy increase is also observed when using a larger Vision Transformer (Dosovitskiy, 2020) backbone.

**Limitations.** The proposed synthetic dataset is effective for general FR but would be less useful for fine-grained tasks such as discriminating between doppelgangers, see Table 6. We believe a better understanding of 'identity' and stronger generation models will help solve this issue and we hope to address this in future work.

## 5 CONCLUSIONS

This paper proposes Vec2Face, a face generation method which can elegantly create a large number of identities and face images. This method conveniently converts vectors to face images and can translate perturbations of the vectors into consistent perturbations of the resulting face images. Because of its unique design, the inter-class and intra-class variations of generated face images can be directly controlled by hyperparameters of vector sampling.. We show that the resulting dataset, HSFace10k, has large intra- and inter-class variations and yields state-of-the-art FR accuracy. Importantly, Vec2Face allows for scaling HSFace to 300K identities and 15M images while still seeing accuracy improvement. In future work, we will study the definition of identity to solve more challenging FR tasks, try other generation structures, and extend this method to generic objects.

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
