# Vec2Face: Scaling Face Dataset Generation with Loosely Constrained Vectors

**Haiyu Wu[1], Jaskirat Singh[2], Sicong Tian[3], Liang Zheng[2], Kevin W. Bowyer[1]**
[1]University of Notre Dame, [2]The Australian National University, [3]Indiana University South Bend
{hwu6@nd.edu, jaskirat.singh@anu.edu, tiansic@iu.edu}
{liang.zheng@anu.edu, kwb@nd.edu}

## A  Appendix

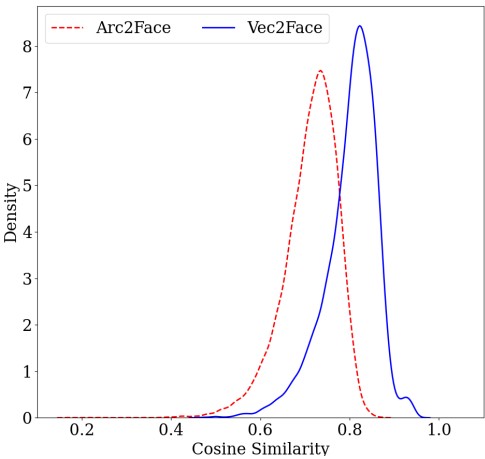

Figure 1: Similarity measurement between original and reconstructed in-the-wild images (LFW (Huang et al., 2008)).

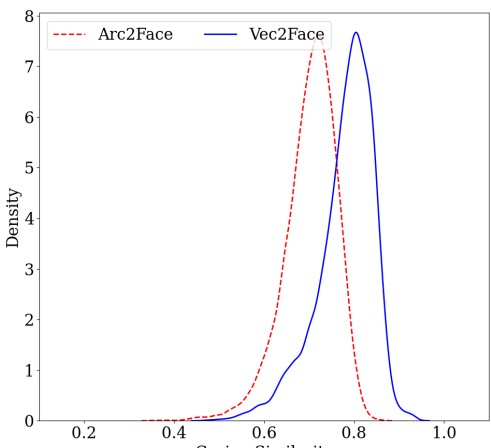

Figure 2: Similarity measurement between original and reconstructed Hadrian images (Wu et al., 2024).

### A.1  Comparison with diffusion model

Training a diffusion model conditioned with image feature vectors is a similar approach to Vec2Face. Thus, we evaluate both architectures by measuring the performance on open-set image reconstruction. For model choice, Arc2Face (Papantoniou et al., 2024) model, which fine-tunes the stable-diffusion-v1-5 on WebFace42M, is used to represent the diffusion models. For test set choices, the images in an in-the-wild dataset, LFW (Huang et al., 2008), and an indoor dataset, Hadrian (Wu et al., 2024) are used. Since both model use face feature as input, we extract image features by using a FR model and feed to both models for image reconstruction. **Because both dataset are trained/fine-tuned on the cropped and aligned FR datasets, no image cropping and alignment are used in this evaluation**. Fig. 1 and Fig. 2 show the similarity distribution between the original images and reconstructed images. The observations are: 1) both models can preserve the identity for the in-door images, 2) Vec2Face has better performance on open-set image reconstruction on both datasets, 3) the diffusion model conditioned with image/identity features do not always preserve the identity, especially for in-the-wild images. Since both model use the features of the same set of images and the variation of the images generated by Arc2Face are only dependent on the initial noise image, we speculate the failed cases of identity preservation are caused by the initial noise image. Hence, the proposed architecture is better than the conventional diffusion model conditioned with image features on identity preservation. Examples are shown in Fig. 11.

## A.2 Ablation study of loss functions

This section presents the effects of identity loss $L_{id}$, perceptual loss, $L_{lpips}$ and GAN loss $L_{GAN}$ on image reconstruction. The observations in Fig. 3 are: 1) without perceptual loss, the reconstructed face edges are smoothed, 2) involving GAN loss at an early training stage causes a glitch effect on image reconstruction, 3) without identity loss, Vec2Face performs well on near-frontal images but not on images with large pose variations.

## A.3 Synthetic FR dataset noise analysis

Dataset noise could be categorized as: i) images from different identities are in the same identity folder, and ii) images from the same identity are in different identity folders. For synthetic datasets, there is no ground truth for identities, so we used the labels provided by the corresponding work.

Following the metric used in previous works (Cao et al., 2018; Wu & Bowyer, 2023; Zhu et al., 2023; Deng et al., 2022), we evaluate the dataset noise using two hard threshold values: one for detecting the outliers within each folder (intra-class noise), and another for detecting similar identities but marked as different in the dataset (inter-class noise). Figure 5 shows the cosine similarity distributions of the available synthetic datasets in situation one. Since greater similarity means images are more likely from the same identity, the results show that our dataset has the highest identity consistency within each identity folder. Moreover, if the similarity value of an image feature and its identity feature is less than 0.3, it can be regarded as an outlier/noise (Deng et al., 2022). With this metric, the available synthetic datasets contain greater intra-class noise than ours. Figure 5 shows the similarity distributions of available synthetic datasets in situation two. WebFace260M (Zhu et al., 2023) is the only work discussing the details of inter-class denoising. It merged two identity folders if their identities have higher than 0.7 similarity, so if the identity similarity is larger than 0.7, we regard it as an inter-class noise. The results show that, except for DCFace and ours, the other datasets have inter-class noise to some extent.

Generally, our proposed dataset has the lowest intra-class noise and inter-class noise. Moreover, Figure 4 shows that the proposed dataset has the lowest overlapped area / Equal Error Rate (EER), showing the best dataset quality on separability between genuine (same people) and impostor (different people) pairs. It promises more reliable identity labels for the FR algorithm to learn better representations.

## A.4 The effect of age and pose variation of datasets in FR accuracy

To estimate the distribution of age and pose, we use img2pose (Albiero et al., 2021) and an age estimator (Albiero et al., 2020) to obtain the data. Fig. 6 shows the distributions of pose and age of five available synthetic datasets and HSFace10k. Since roll is controlled by face detectors, only pitch and yaw angles are estimated. The results show that all the datasets have large variations on both pose and age, suggesting the accuracy difference is not mainly caused by the attribute variation but by identity consistency. Having large variations but losing the identity consistency results in lower accuracy.

## A.5 Feature interpolation and feature value impact on generated images

We analyze the impact of feature values on the generated images in two ways: 1) interpolating six feature vectors between two image features and 2) proportionally changing the values in the feature vector. Figure 7 shows the results of feature interpolation between two images. It indicates that the interpolation in the feature domain can be smoothly presented in the image domain, showcasing Vec2Face's capability of understanding the vector characteristics in the feature domain. Figure 8 shows the results of proportionally changing the values in the feature. The observations are: 1) As the values in a feature vector increase to a large extent, the image stops changing but the quality decreases, for both negative and positive directions; 2) As the values in a feature vector become close to zero, the face attributes are erased and eventually disappeared.

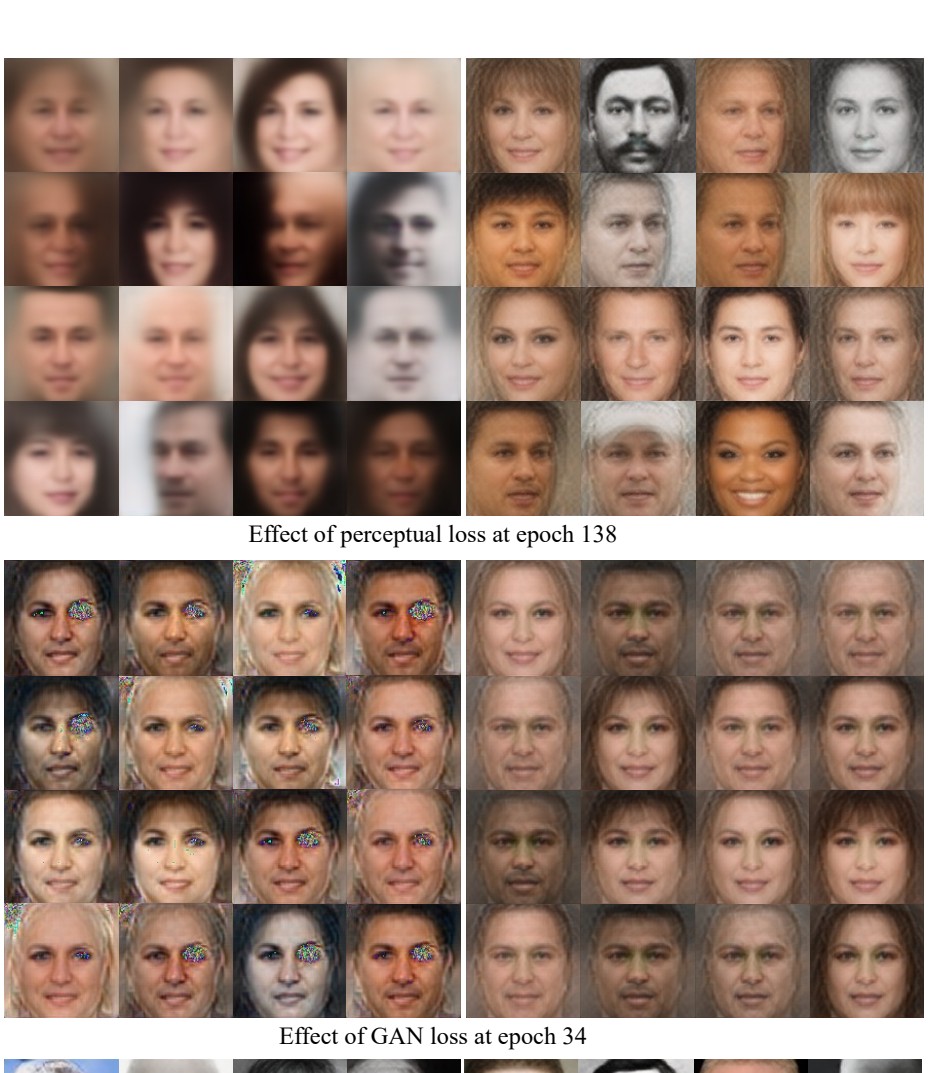

Effect of perceptual loss at epoch 138

Effect of GAN loss at epoch 34

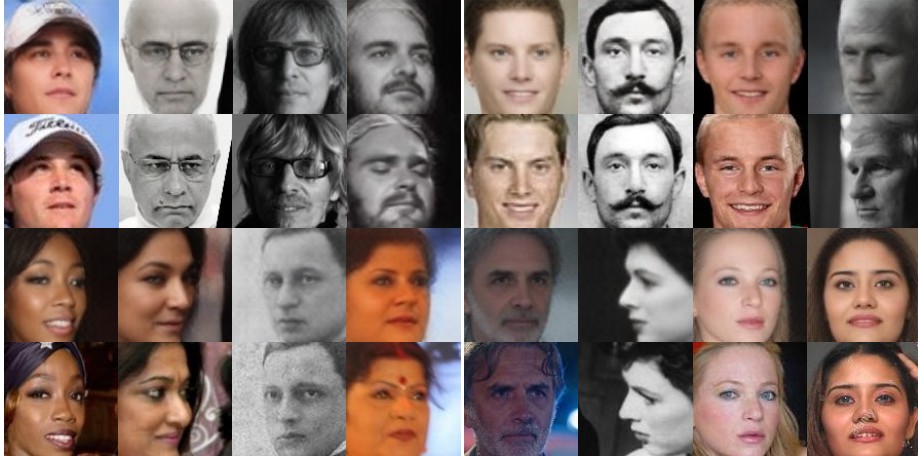

Effect of identity loss

Figure 3: Examples of loss effect during training. The reconstructed examples with corresponding loss involved are on the left, otherwise on the right. For identity loss, odd rows are reconstructed images and even rows are original images.

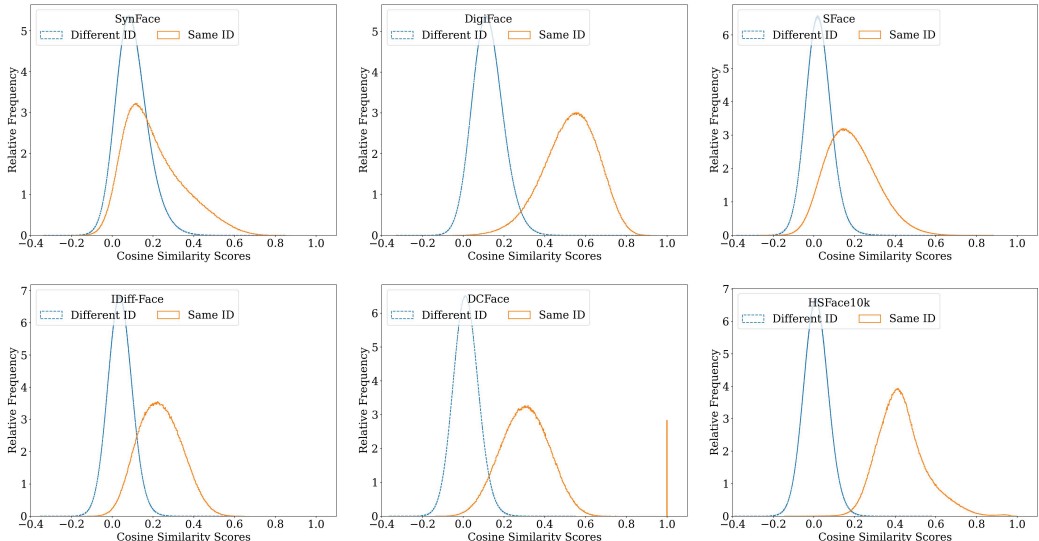

Figure 4: Similarity distributions of available synthetic datasets (Qiu et al., 2021; Boutros et al., 2022; Bae et al., 2023; Boutros et al., 2023; Kim et al., 2023) and ours. The results indicate that our dataset has the largest separability and smallest overlap / Equal Error Rate between genuine (**same ID**) and impostor (**different ID**) pairs.

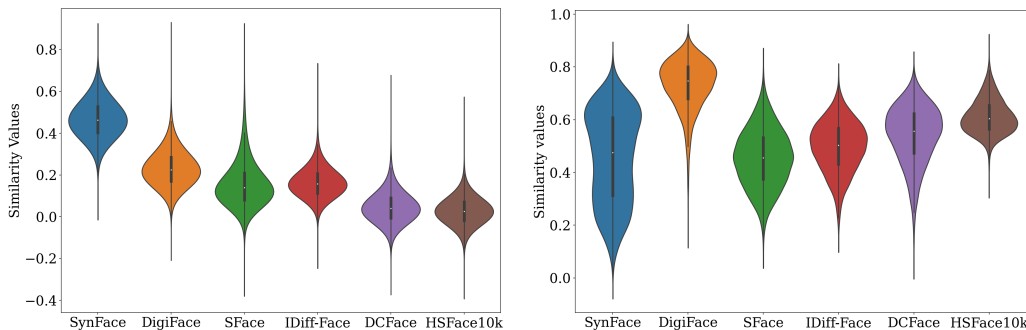

Figure 5: The inter-class and intra-class similarity distributions of the available synthetic datasets (Qiu et al., 2021; Boutros et al., 2023; Bae et al., 2023; Kim et al., 2023; Boutros et al., 2022) and ours.

## A.6    IMPACT OF VALUES AT DIMENSIONS

Vec2Face generates an image based on the values in a feature vector, which motivates an intriguing investigation of how values in dimensions impact the output image. We start at changing the value in one dimension. Fig. 13 shows the cases where some obvious patterns occurring. For each of the four identities, the value at index 15 is related to the hair volume, pose angle, age and expression; the value at index 26 have a high correlation with the bangs, facial hair and age, eyeglasses presence, and expression; increasing the value at index 32 changes the age, head pose, hair color, and expression. The observation is that changing values in a single dimension may change the face attribute but it is not consistent for all identities. In fact, changing value at a single dimension does a negligible change on the generated image in most dimensions.

We then change the values in fixed-length (*i.e.*, 8) of dimension chunks and there are some noticeable patterns. Fig. 12 shows the generated images when the values at specific dimensions increases. A general conclusion is that changing values at dimension chunks varies the facial attributes but the patterns are also inconsistent across identities. For instance, dimension [40:48] changes the age for the first identity, no obvious pattern for the second, the expression for the third, and head pose for the fourth; dimension [56:64] changes the hairstyle for the first identity, facial hair for the second, age and eyeglasses for the third, age and facial hair for the fourth; dimension [96:104] changes the gender

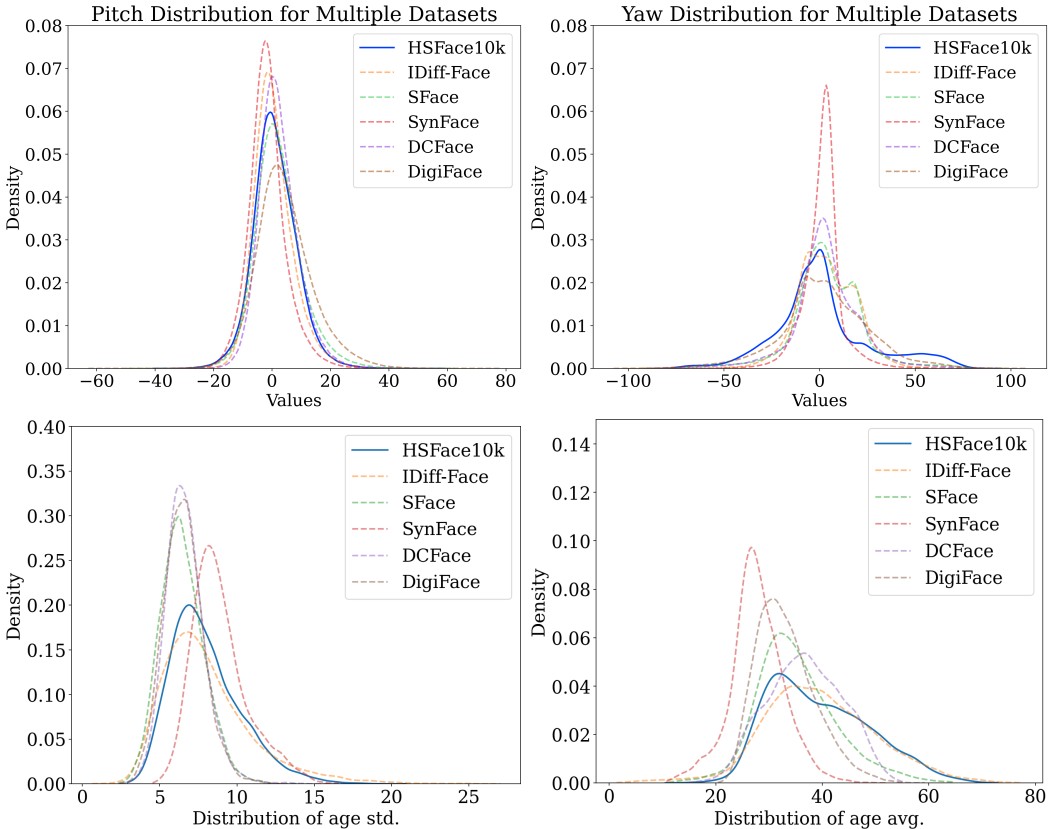

Figure 6: Pose and age variation across five test sets and HSFace10k (ours).

for the first identity, age and hairstyle for the second, face exposure level for the third, and facial hair for the fourth. Therefore, it is hard to control the attributes by handcrafting the features. AttrOP is much more efficient and effective.

## A.7 DISCUSSION OF PRIVACY ISSUE AND FUTURE DATASET USAGE

The privacy issue of using images from real identities is the main concern in face recognition technique development. As a result, governments publish regulations (Voigt & Von dem Bussche, 2017) to restrict biometric data usage. Before the possible fully restrictions on real face data usage, although not achieving as good performance as using datasets of real data (see in Table 1), this work provides several datasets for continuing the development of face recognition techniques without violating the regulations. Moreover, the unique design of Vec2Face can be inspired for generating better synthetic datasets in the future.

| Datasets | HSFace300K | WebFace4M (50K) | Glint360K |
|---|---|---|---|
| Accuracy | 93.52 | 96.61 | 97.63 |

Table 1: FR accuracy comparison of the model trained with the largest proposed data HSFace300K, the 50K identities used for Vec2Face training, and the pretrained FR model used for feature extraction.

## A.8 IDENTITY LEAKAGE

To validate the efficacy of Vec2Face in generating a large number of synthetic identities without causing identity leakage, we conduct two experiments: 1) measuring the identity leakage between identities in the synthetic dataset and WebFace4M identities, and 2) measuring the identity leakage between randomly generated identities and identities in two real datasets. In (Zhu et al., 2023), two identities that have a similarity higher than 0.7 is regarded as the same identity. We use a pre-trained FR model to extract the face feature and adopt the same threshold to measure the identity leakage in percentage. For the former, Table 3 shows that both DCFace and HSFace10K do not have identity leakage issues, but CemiFace has a 24.21% identity leakage. Hence, DCFace and HSFace10K do not

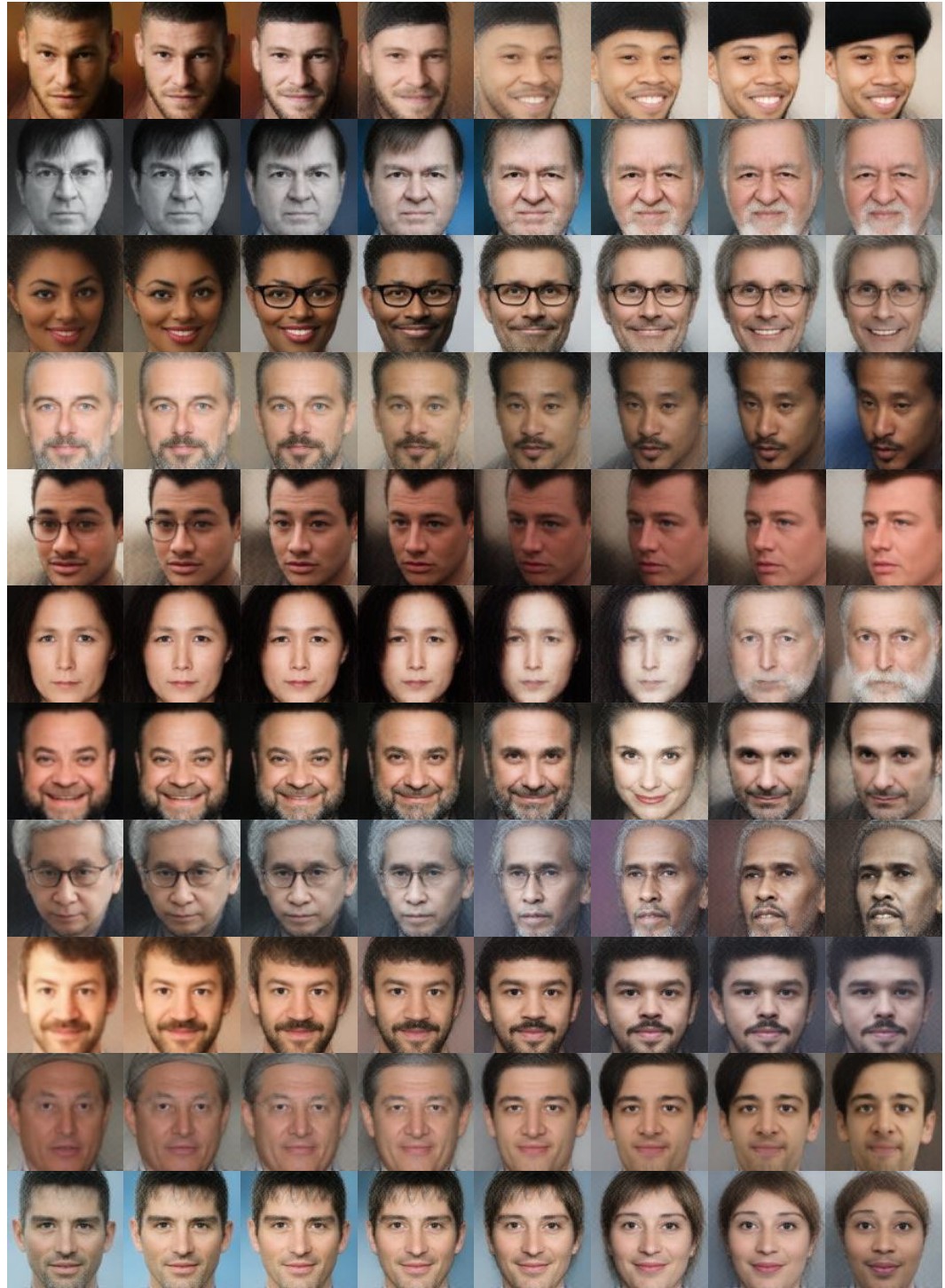

Figure 7: Feature interpolation results. These show that Vec2Face can smoothly convert one image to another by simply changing the feature values.

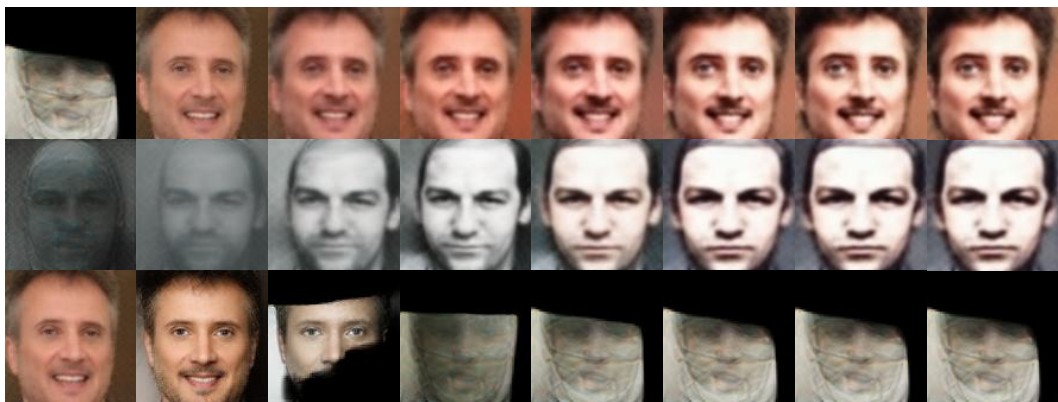

Figure 8: Impact of feature values analysis. The first row shows images generated with feature values, from left to right, of {0, 1, 1.5, 2, 3, 10, 100, 200}. The second row shows images generated with feature values of {-0.5, -1, -1.5, -2, -3, -10, -100, -200}. The last row shows images generated with feature values of {1, 0.5, 0.25, 0.125, 0.0625, 0.03125, 0.015625, 0}.

| Real + synthetic | Avg. |
| --- | --- |
| CASIA-WebFace | 94.79 |
| + HSFace20K | 95.67 |
| + HSFace100K | 95.69 |
| + HSFace200K | 95.43 |
| + HSFace300K | 95.72 |
| + HSFace400K | 95.64 |

Table 2: Average accuracy of the FR model trained with mixed training sets (real + synthetic) tested on LFW, CFP-FP, AgeDB-30, CALFW, and CPLFW.

| Dataset | ID leakage (%) |
| --- | --- |
| DCFace | 0% |
| HSFace10K | 0% |
| CemiFace | 24.21% |

Table 3: Identity leakage of DCFace, CemiFace (Sun et al., 2024), HSFace10K on WebFace4M dataset. Following the strategy in WebFace4M, we calculate the percentage of synthetic IDs that have a similarity larger than 0.7 with the WebFace4M IDs. CemiFace has a large percent ID leakage.

| 5M random IDs | 0.5 | 0.7 |
| --- | --- | --- |
| CASIA-WebFace | 0% | 0% |
| WebFace4M | 0% | 0% |

Table 4: ID leakage on 5M randomly sampled IDs. Using the same strategy in Table 3, the results indicate that the Vec2Face model has a huge potential to generate a large number of synthetic identities without causing ID leakage issues.

violate the regulations, whereas CemiFace does. For the latter, Table 4 indicates that even though we randomly generate 5M identities, Vec2Face does not cause any ID leakage issue, showcasing the potential of effectiveness of synthetic identity generation and dataset scalability.

## A.9 OTHER MATERIAL

The performance of mixing real and synthetic training sets is in Table 2. The architecture of fMAE is in Figure 10. The FR training configuration is in Table 5. The examples of the effect of stochasticity in fMAE on generated images is in Figure 9.

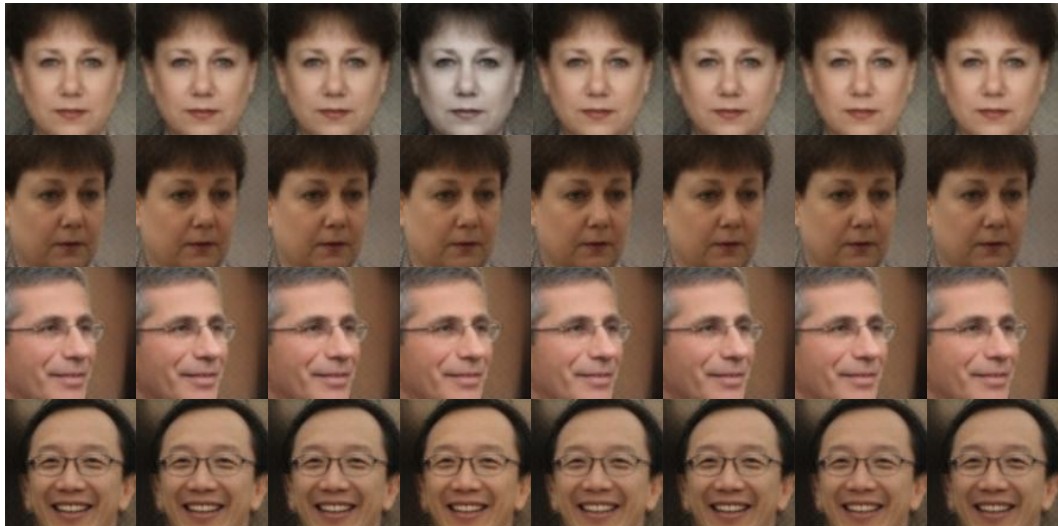

Figure 9: Examples of the effect of the stochasticity of fMAE. Each vectors is processed 8 times for image generation. The changes on images are negligible.

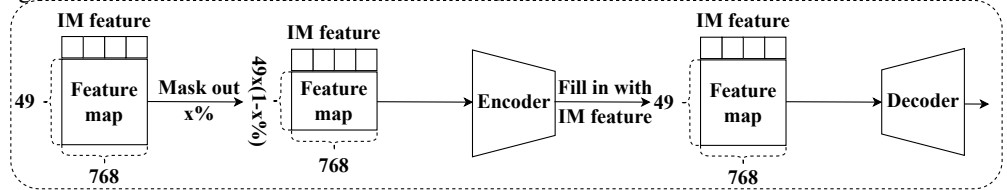

Figure 10: Architecture of the proposed feature masked autoencoder (fMAE).

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

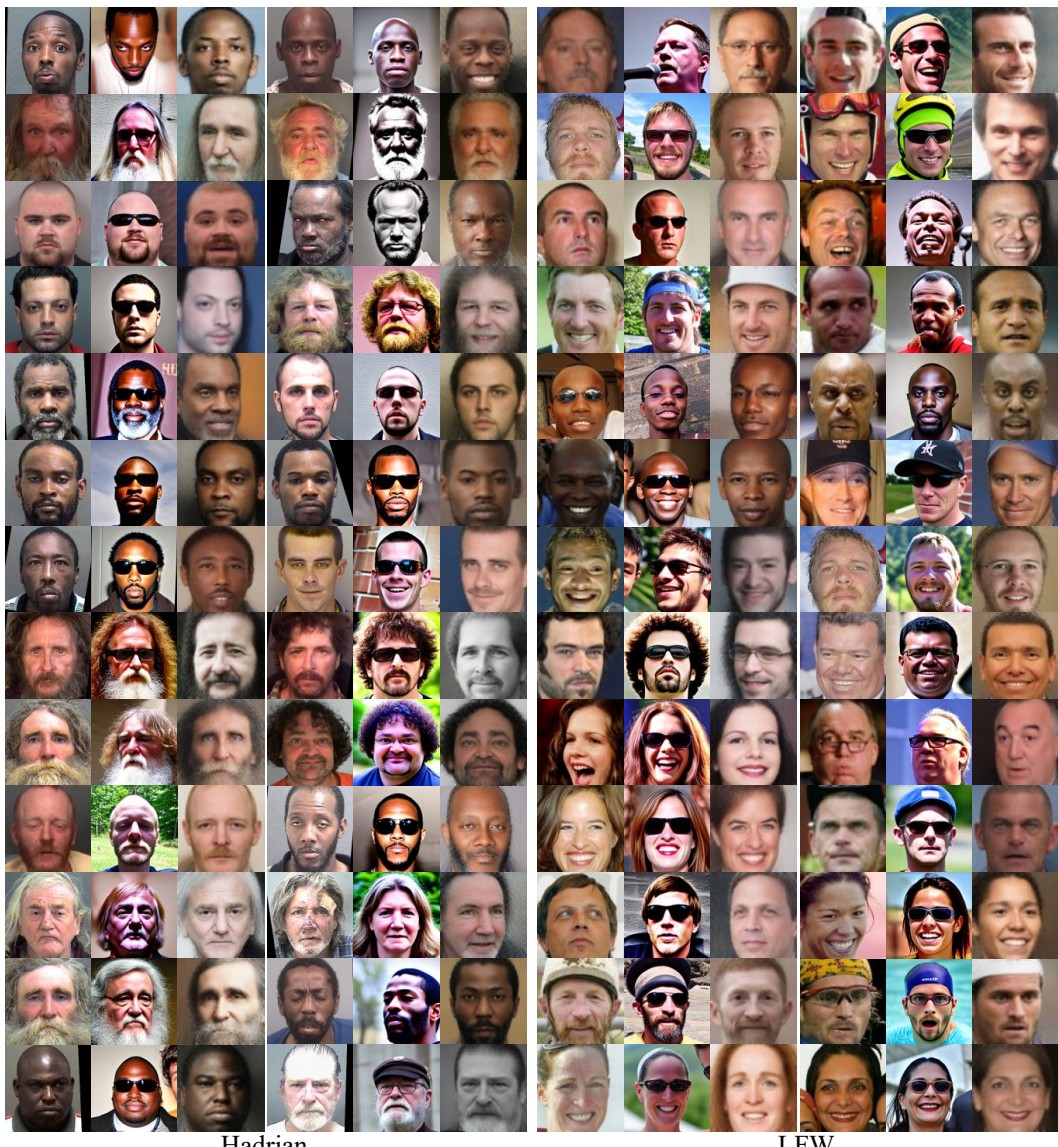

Hadrian                                    LFW

Figure 11: Examples of reconstructed Hadrian and LFW images by Arc2Face and Vec2Face. The images in each three-image group, from left to right, are the original, from Arc2Face, and from Vec2Face.

Changes happen in [40:48]

Changes happen in [56:64]

Changes happen in [96:104]

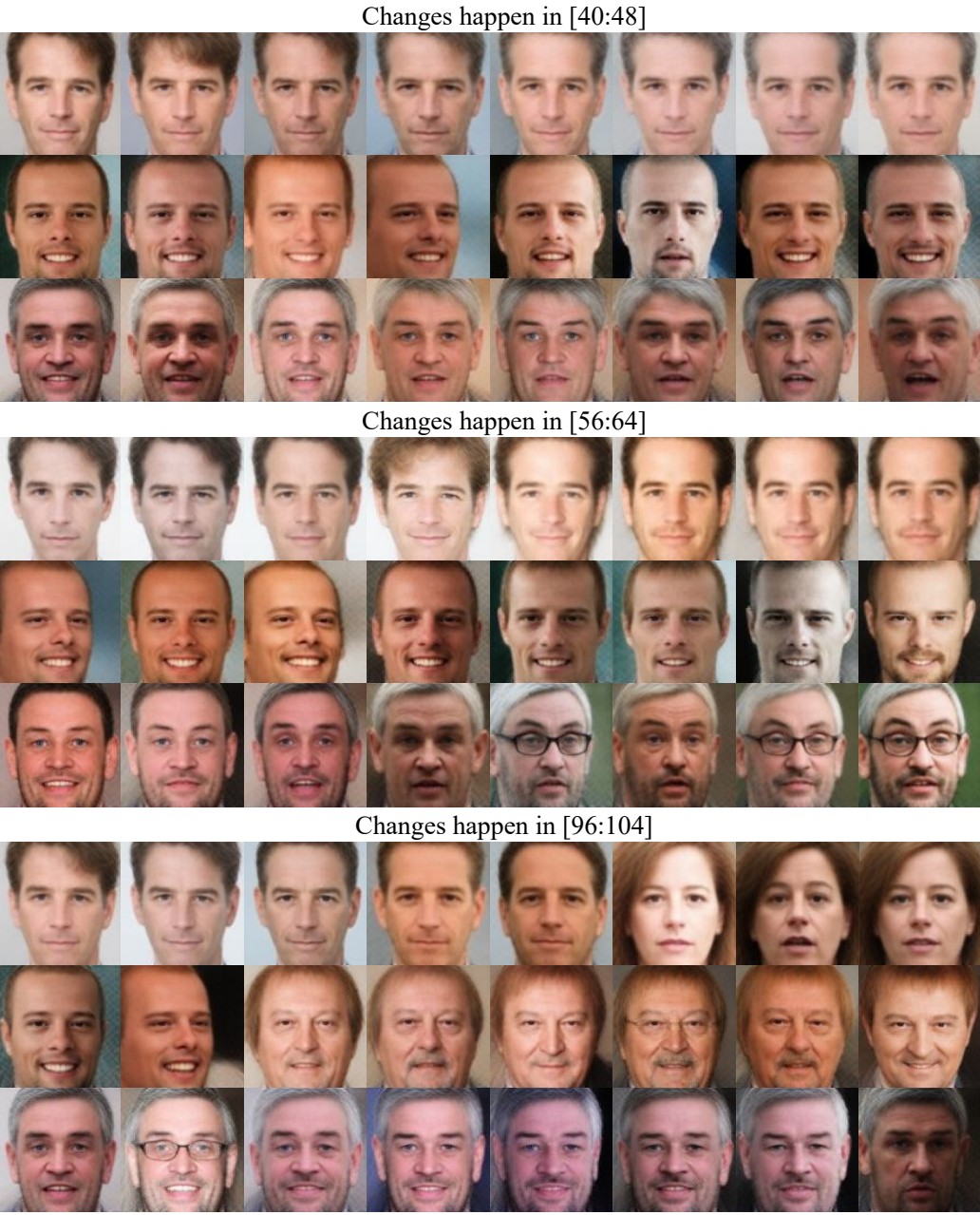

Figure 12: Examples of changing values in a single dimension. From left to right, the value gradually increases in the target dimensions. The raw images are in low quality, so we use AttrOP to increase the image quality.

Changes happen in [15]

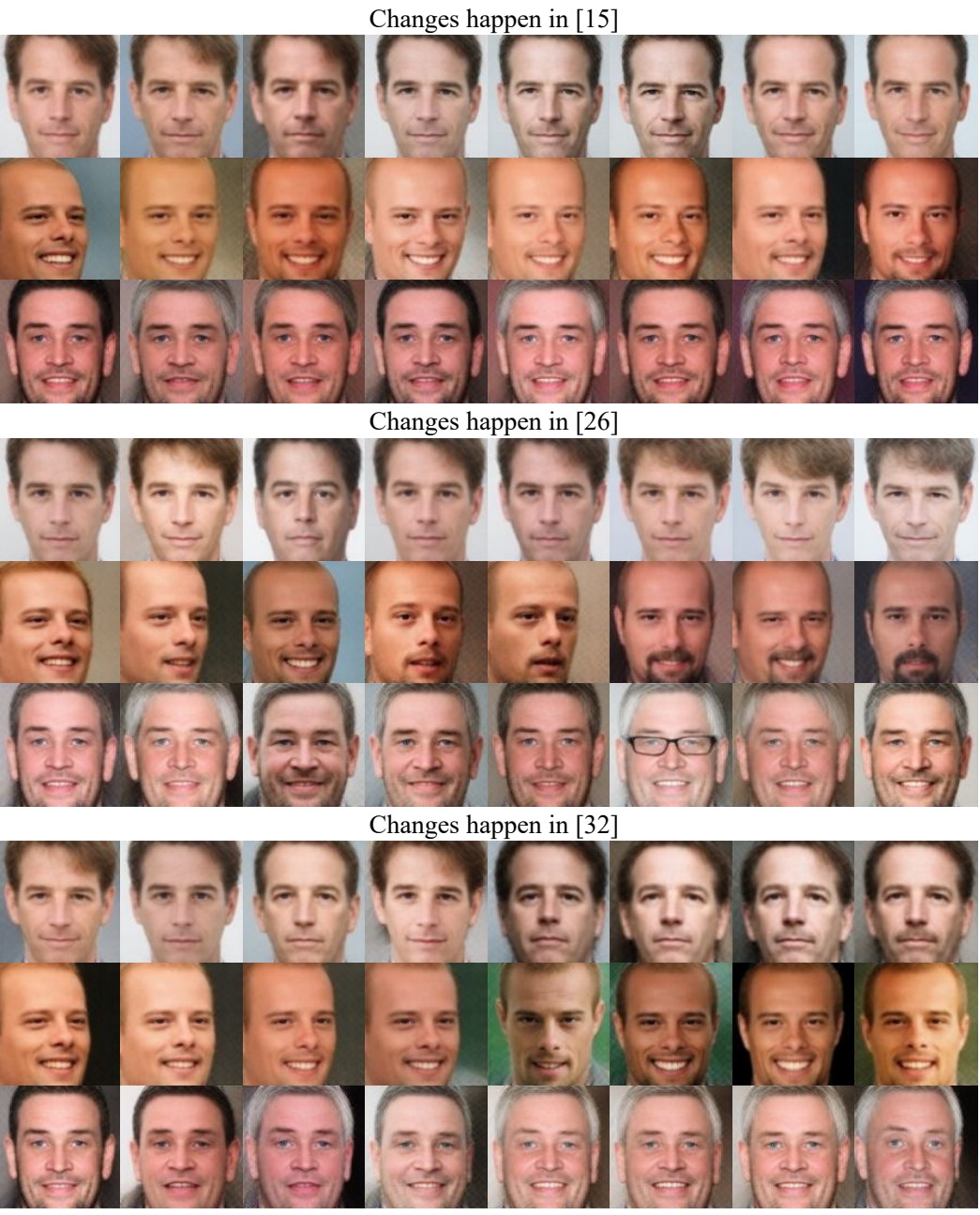

Changes happen in [26]

Changes happen in [32]

Figure 13: Examples of changing values in a single dimension. From left to right, the value gradually increases in the target dimensions. The raw images are in low quality, so we use AttrOP to increase the image quality.

| Face Recognition Model Training Configurations | |
|---|---|
| Head | ArcFace |
| Backbone | SE-IR50 |
| Input Size | 112×112 |
| Batch Size | 128 |
| Learning Rate | 0.1 |
| Weight Decay | 5e-4 |
| Momentum | 0.9 |
| Epochs | 26 |
| Margin | 0.5 |
| FP16 | True |
| Sample Rate | 1.0 |
| Reduce Learning Rate | [12, 20, 24] |
| Augmentation | Random aug. and Random erase |
| Optimizer | SGD |
| Workers | 2 |
| GPU | RTX6000 |

Table 5: Configurations used for synthetic dataset training.

Foivos Paraperas Papantoniou, Alexandros Lattas, Stylianos Moschoglou, Jiankang Deng, Bernhard Kainz, and Stefanos Zafeiriou. Arc2face: A foundation model of human faces. *arXiv preprint arXiv:2403.11641*, 2024.

Haibo Qiu, Baosheng Yu, Dihong Gong, Zhifeng Li, Wei Liu, and Dacheng Tao. Synface: Face recognition with synthetic data. In *ICCV*, pp. 10860–10870, 2021.

Zhonglin Sun, Siyang Song, Ioannis Patras, and Georgios Tzimiropoulos. Cemiface: Center-based semi-hard synthetic face generation for face recognition. *NeurIPS*, 2024.

Paul Voigt and Axel Von dem Bussche. The eu general data protection regulation (gdpr). *A Practical Guide, 1st Ed., Cham: Springer International Publishing*, pp. 10–5555, 2017.

Haiyu Wu and Kevin W Bowyer. What should be balanced in a" balanced" face recognition dataset. In *BMVC*, pp. 2, 2023.

Haiyu Wu, Sicong Tian, Aman Bhatta, Jacob Gutierrez, Grace Bezold, Genesis Argueta, Karl Ricanek Jr., Michael C. King, and Kevin W. Bowyer. What is a goldilocks face verification test set? *arXiv preprint arXiv:2405.15965*, 2024.

Zheng Zhu, Guan Huang, Jiankang Deng, Yun Ye, Junjie Huang, Xinze Chen, Jiagang Zhu, Tian Yang, Dalong Du, Jiwen Lu, and Jie Zhou. Webface260m: A benchmark for million-scale deep face recognition. *TPAMI*, pp. 2627–2644, 2023.