# OpenReview forum: "Vec2Face: Scaling Face Dataset Generation with Loosely Constrained Vectors"
_ICLR.cc/2025/Conference — ICLR 2025 Poster_

### Official Review · Reviewer_LGwK · 2024-10-30

**Soundness:** 3
**Presentation:** 3
**Contribution:** 2
**Rating:** 6
**Confidence:** 5

**Summary:**

This paper proposes VEC2FACE to synthesize facial images from a sampled identity vector for synthetic face recognition. It utilizes a GAN solution to reconstruct the image at the training stage and adopt the decoder for inference. High intra-class variation is ensured by Perturbing identity vectors. comprehensive experiments are conducted to demonstrate the effectiveness of the method.

**Strengths:**

1. The proposed method achieves good performance compared to the listed literature.
2. It can generate as many novel identities as possible.
3. The experiments are sufficient.
4. Using GAN to achieve such results is inspiring.

**Weaknesses:**

1. [Major] lines 219-220, the authors state 'Following (Papantoniou et al., 2024), PCA is used to sample vectors.'. This brings a concern that the contribution of generating as many novel identities as possible is benefited from the previous work Arc2Face. Additionally, adopting pure identity embedding to generate a facial dataset is demonstrated in the related literature, ID3.

2. [Major] After attribute OP, how do you make sure the generated image preserves the identity, Do you have experiments to prove it? The attribute operation is also introduced in ID3.

3. [Major] line 314, Vec2Face training set is WebFace4M, the PCA is carried on MS1MV2, the feature extraction model is from Glint360K. This counters the privacy concerns, as the dataset original MS1M dataset is recalled by its creator. And the whole training requires a lot of face recognition dataset, which is a disadvantage over DCFace/ID3.

4. [Major] In addition, the training dataset is webface4m (1M with 50k identities ), which is significantly larger than CASIA-WebFace (500k images with 10k identities) adopted by the competitors DCFace/ID3, making the comparison with previous methods unfair.

5. [Major] The authors state they achieve state-of-the-art results. However, the SOTA result is 92.30 (on 0.5M volume) and 93.07  (on 0.5M volume), achieved by CemiFace[1].

6. [Minor]How to generate the image is not clear. For example, how do you obtain the identity feature, randomly generated or from the image?

7. [Minor] A study about alternative pretrained FR model should be conducted, as the proposed method uses ArcFace pretrained from Glint360k, DCFace adopted model pretrained from WebFace4M.

8. [Minor] Line 243, ‘too few profile head poses’. Do you have statics results?

9. [Minor] Eq. 5 is trained with the GAN?

10. Long-page discussions might be better if positioned in the appendix.

Reference:
CemiFace: Center-based Semi-hard Synthetic Face Generation for Face Recognition

**Questions:**

Please see the weakness.

Overall, I believe the novelty of this paper is incremental, and improvement is good but does not achieve state-of-the-art. Furthermore, privacy issues arise as this paper adopts a lot of labelled face recognition datasets.

---

> ### Author Response · Authors · 2024-11-19
>
> We appreciate your comments and pointing out two new papers. **However, we want to emphasize that ID3 and CemiFace were released on arXiv (not officially published) on Sep. 26 and Sep. 27, which was close to the ICLR abstract submission deadline (Sep. 27) and the full paper submission deadline (Oct. 1). Hence, we think it is unfair to either compare our paper with these two or suppose the algorithms in our paper were borrowed from these two papers.**
>
> ## Answers of Weaknesses
>
> ### Weakness 1.
> The proposed model, Vec2Face, which can convert vectors to images while preserving the ID information, is the reason we can generate an unlimited number of identities, not PCA. The reason we use PCA is that we found it can sample better/stronger identity vectors, resulting in accuracy improvement.
> |       Vector sampling          | Accuracy |
> | :-------------: | :------: |
> |  with PCA   |  92.00   |
> | without PCA |  91.24   |
>
> Unlike Arc2Face, which fine-tunes a stable-diffusion-v1.5 on WebFace42M, we train our model with only 1M images and we can achieve higher accuracy, showcasing the efficacy of the proposed work.
>
> As for ID3, we added its result because we saw that literature before the full version submission. We will add it to the related work, but it is impossible for us to borrow ideas from it because we trained our model for over 1000 epochs and the HSFace400K has 20M images. Both are non-trivial to be achieved in this short period.
>
> ### Weakness 2
> Yes, we do. The genuine distribution in Figure 4 and intra-class similarity distributions in Figure 5 of the Appendix show that the average intra-class similarity of HSFace10K is better than most of the prior synthetic datasets. Moreover, a pre-trained FR model (M_{FR}) is used in Algorithm 1 and Equation 5 to ensure that the output of AttrOP is identity-preserving.
>
> ### Weakness 3
> We use three different datasets due to privacy concerns reported in [1]. If we use WebFace4M identities to learn PCA, it might be easier for Vec2Face to generate identities that are close to the identities in WebFace4M due to the leakage. In fact, the results below show that using WebFace4M to learn PCA does benefit the accuracy, albeit incrementally. As for using the FR model trained with Glint360K, we want to use the SOTA FR model to generate good image embeddings. It can be any other model or a special design to map vectors to images with a well-defined pattern.
>
> |                 | Accuracy |
> | :-------------: | :------: |
> |  PCA on MS1MV2   |  92.00   |
> | PCA on WebFace4M |  92.05   |
>
> ### Weakness 4
> We retrained the Vec2Face model with CASIA-WebFace, generated a new HSFace10K dataset, and the accuracy is shown below.
>
> |                 | Accuracy |
> | :-------------: | :------: |
> |  DCFace   |  89.56   |
> |HSFace10K | **89.67** |
> | ID3 |  89.80   |
>
> We use 1M images because we found that 0.5M images are insufficient for the ViT-based architecture, as the ViT model is known to be data-consuming [2]. Specifically, when using randomly sampled vectors (sampled from N(0, 1), not from PCA), the images are not well-structured. With an underfit model, it can still outperform DCFace, showing the efficacy of the proposed algorithm.
>
> ### Weakness 5
> Thanks for pointing out this paper to us. Their study on choosing the correct similarity range for image generation is interesting and inspiring. This was the headache when we tried to find a good \sigma combination for intra-class vector sampling. However, based on the analysis in DCFace, we don't think they can generate an unlimited number of well-separated identities (80K at most). The analysis of Vec2Face vs. Arc2Face shows that even with identity conditions, it is still hard for the diffusion model to generate well-separated identities. Since Vec2Face allows us to easily scale the dataset, our accuracy of 93.52% with 300K identities remains the SOTA performance.
>
> We believe combining the core idea of Vec2Face and CemiFace can be worth trying. **This will be explored in our future work.**
>
> ###  Weakness 6
> Vec2Face generates images from vectors, so the input is a vector. Identity vectors are randomly sampled from the learned PCA. The pipeline is: 1) randomly generating identity vectors, 2) perturbing sampled identity vectors, and 3) feeding these vectors to Vec2Face to obtain images. For AttrOP, the input vectors are from step 2).
>
> ### Weakness 7
> We will use the FR model trained with WebFace4M to measure the identity separability, but it should not change the observation.
>
> ### Weakness 8
> We will add it to the Appendix.
>
> ### Weakness 9
> Eq. 5 is used in AttrOP (Algorithm 1), which is used at the inference stage. The training loss is Eq. 3.
>
> ## Answers of Questions
> The labeled face datasets are not a must because we only use the features and images, not the labels.
>
> Reference: \
> [1] This Face Does Not Exist... But It Might Be Yours! Identity Leakage in Generative Models \
> [2] An Image is Worth 16x16 Words: Transformers for Image Recognition at Scale

---

> ### Author Response · Authors · 2024-11-25
>
> Dear Reviewer LGwK,
>
> We hope your work goes smoothly, and thank you for pointing these two papers out again.
>
> We admit that these two works report comparable accuracy (0.13% and 0.3% higher) to ours, but the highest accuracy (93.52%) we achieved in this paper is still state-of-the-art, due to the proposed method allowing us to generate an **unlimited number of well-separated** synthetic identities, which is not applicable to these two new papers. This is a big breakthrough in this area and can be combined with any other novel methods, e.g., CemiFace, to generate a better synthetic dataset.
>
> Although it is unfair to compare with these two papers, we have put a lot of effort into addressing your concerns and questions.
>
> We believe that the proposed method and the achieved functionality are still valuable to the field and genuinely hope you can reconsider this paper's quality and contributions.

---

> > ### Comment · Reviewer_LGwK · 2024-11-26
> >
> > Thank you to the authors for their thoughtful responses. They have addressed some of my concerns; however, I still believe there are issues that need further clarification:
> >
> > (1) ViT vs. ResNet:
> > Why was ViT chosen instead of ResNet? This choice seems to make the comparison less fair. To ensure a more balanced evaluation, I suggest using a pretrained ViT, such as ViT pretrained with DINO(Self-supervised learning).
> >
> > (2) Privacy Leakage:
> > I agree with Reviewer G33d that there are potential privacy leakage concerns that have not been adequately addressed.
> >
> > (3) ID3 Benchmarking:
> > I noticed the inclusion of recent papers, but the authors have excluded some with better performance. For instance, Table 1 assumes ID3Face is comparable, but it omits CemiFace, which appears relevant.
> >
> > (4) Unlimited Identities:
> > The argument about supporting 400K identities is unconvincing. For example, DCFace could handle up to 2M identities by sampling one-shot data from WebFace42M, which contains 2M identities.
> >
> > Overall, although I believe this paper achieved good performance, however, there are still some problems that need to be addressed in the revision.

---

> > > ### Author Response · Authors · 2024-11-27
> > >
> > > ## ViT vs. ResNet
> > > We are not sure why ViT vs. ResNet makes the comparison less fair, because we only list the FR accuracy of the ViT and ResNet models trained with the proposed HSFace10K in Table 9 of the main paper. This is to show the proposed dataset can be used to train both the ViT backbone and ResNet backbone. There is no comparison in that table.
> > >
> > > As for DINO, it is not related to face recognition and none of the previous works use it. While it might be interesting to see what the performance would be, it is not within the scope of this work.
> > >
> > > The only component in the proposed model that is related to ViT is the fMAE. We simply follow the default setting of the original MAE.
> > >
> > > We mentioned ViT in the previous answer because there was a question about why we use 1M images instead of 0.5M to train the model. To fairly compare with other methods, we conducted a further experiment
> > >
> > > The HSFace datasets in this table are generated by the Vec2Face model retrained on CASIA-WebFace. At the 0.5M scale, we do not achieve higher accuracy than ID3. However, since our method allows us to scale the dataset up, we scaled the identities to 80K, where the final accuracy is significantly higher than ID3.
> > >
> > > |            | Accuracy  |
> > > | :--------: | :-------: |
> > > |   DCFace   |   89.56   |
> > > |    ID3     |   89.80   |
> > > | HSFace10K  |   89.67   |
> > > | HSFace80K | **92.77** |
> > >
> > > ## Privacy Leakage
> > > We use the Vec2Face model trained with CASIA-WebFace to generate 5M identity images and compare these identities with the WebFace4M identities and CASIA-WebFace identities. This table shows that the overlap is small and can be easily fixed by removing these overlapping identities. Thus, potential privacy leakage can exist but can be easily fixed due to the advantage of the proposed method. Experiment details can be found under Reviewer oM4G.
> > >
> > > | 5M randomly sampled IDs |  0.4  | 0.6  | 0.7  |
> > > | :---------------------: | :---: | :--: | :--: |
> > > |        WebFace4M        | 1.27% |  0   |  0   |
> > > |      CASIA-WebFace      | 0.17% |  0   |  0   |
> > >
> > > ## ID3 Benchmarking and Unlimited Identities
> > > Again, we thank you for mentioning these two papers. We don't think using this one-shot strategy matches the goal of the synthetic dataset generation, where no real identity should be included in the dataset. The reason is that the one-shot strategy uses images from a real identity which leads to identity leakage. Moreover, DCFace uses a pre-trained image generation model to obtain the identity and only transfers the style of the real images. It is totally different from the one-shot strategy used in the CemiFace paper.
> > >
> > > Since CemiFace has released its datasets, we conducted two simple experiments to evaluate the **identity separability** and test the **identity leakage**.
> > >
> > > ### Identity separability
> > > Using the same approach as in Section 4.4, we measure the identity separability of CemiFace. This table shows that CemiFace identities are not as well-separated as ours.
> > > | Datasets      | \# of well-separated IDs / \# of total IDs |
> > > | ------------- | :--------------------------------------: |
> > > | DCFace        |               7363 / 10000               |
> > > | CemiFace      |               8512 / 10000               |
> > > | HSFace        |             **9800** / 10000             |
> > > | CASIA-WebFace |               9842 / 10000               |
> > >
> > > ### Identity leakage
> > > Experiment steps:
> > > 1. We use a pre-trained FR model to calculate the identity features in the datasets below and WebFace4M.
> > > 2. We use FAISS to search for the highest identity match for each identity in the datasets below from WebFace4M
> > > 3. We use three thresholds to evaluate identity leakage by calculating how many identity pairs have similarity values higher than the threshold.
> > > 4. We report the percentages.
> > >
> > > | Datasets\threshold |  0.4   |  0.6   |  0.7   |
> > > | :----------------: | :----: | :----: | :----: |
> > > |   CASIA-WebFace    | 21.03% | 7.93%  | 7.87%  |
> > > |      CemiFace      | 95.22% | 59.36% | 24.21% |
> > > |       DCFace       | 8.42%  |   0    |   0    |
> > > |     HSFace10K      | 14.32% |  8.9%  | 8.88%  |
> > > |     HSFace300K     | 1.49%  |  0.6%  |  0.6%  |
> > >
> > > According to the WebFace4M paper [1], the identity pairs that have a similarity higher than 0.7 can be considered the same identity. Thus, CemiFace has a severe identity leakage problem.
> > >
> > > As for the proposed HSFace datasets, they have identity leakage too, but it can be easily solved by sampling more identities until there is no leakage. Moreover, as we mentioned previously, CemiFace and the proposed work can be combined to achieve a better performance, because CemiFace needs identity images to generate datasets and our work can generate a large number of identities. This showcases the importance of the proposed work. **All analyses and results regarding CemiFace will be added to the paper.**
> > >
> > > As for ID3, the previous section has addressed it.
> > >
> > > Reference:\
> > >  [1] WebFace260M: A Benchmark Unveiling the Power of Million-Scale Deep Face Recognition

---

> > > > ### Comment · Reviewer_LGwK · 2024-11-27
> > > >
> > > > Thank the authors for their feedback. I have raised my score from 5-6.

---

> ### Author Response · Authors · 2024-11-27
>
> We appreciate your decision!

---

### Official Review · Reviewer_G33d · 2024-10-31

**Soundness:** 2
**Presentation:** 3
**Contribution:** 3
**Rating:** 5
**Confidence:** 5

**Summary:**

The paper introduces a novel method called Vec2Face to synthesize large-scale, high-quality face images for training face recognition models. This method addresses the limitations of existing techniques by:
1. Generating a vast number of distinct identities: Vec2Face can generate up to 300,000 unique identities, significantly surpassing previous methods.
2. Controlling intra-class variation: The model can generate diverse variations of the same identity by perturbing the input vector, enabling realistic appearance changes.
3. Manipulating attributes: Vec2Face allows for the generation of images with specific attributes by adjusting the input vector using gradient descent.

**Strengths:**

1. The pipeline presented in the paper is simple and efficient.
2. The use of a feature-masked autoencoder is somewhat novel.

**Weaknesses:**

1. Limited Scale of Synthetic Data: While the paper proposes a method for generating synthetic faces, the number of identities generated (300K-400K) is comparable to existing real-world datasets used to train facial recognition (FR) models. This raises concerns about the practical benefits of generating synthetic data at this scale.
2. Lack of Comparison with Larger Datasets: The paper lacks experiments comparing the effectiveness of the generated synthetic data with larger-scale real-world datasets like Glint360K. This comparison is crucial to demonstrate the advantages of the proposed approach in scenarios where abundant real data is available.
3. Unclear Practical Motivation: The authors do not provide a compelling reason for generating a synthetic dataset with a similar or smaller scale than commonly used real-world FR training datasets.  It is essential to clarify the specific use cases and advantages of this approach, particularly when real data is readily accessible.
4. Missing Citation: The paper omits a key citation: "Vec2face: Unveil human faces from their black box features in face recognition" (CVPR 2020). This work appears relevant to the paper's methodology and should be properly acknowledged.

Misleading Novelty Statements: Certain claims of novelty in the paper are potentially misleading.  (See the next questions section for more details.)  It is crucial to ensure that all claims are accurate and supported by evidence.

**Questions:**

1. Choice of Pre-trained FR Model:
    a. You've chosen to use ArcFace-R100 pre-trained on Glint360K. Could you elaborate on the reasons behind this specific choice?
    b. Are there any limitations to using a smaller pre-trained FR model trained on a smaller dataset? It seems that requiring a large-scale, pre-trained FR model could limit the applicability of your approach, especially in scenarios where such a model isn't readily available. This creates a potential "chicken-and-egg" problem where generating synthetic data requires access to large-scale real data. Could you address this potential limitation?

2. Number of Identities:
    a. Is there a specific reason for limiting the number of generated identities to 300K or 400K?
    b. Given that your filtering process removes only 1.7% of generated images, is it possible to generate a larger number of identities (e.g., 500K or 1M)? If so, what are the potential challenges or trade-offs involved?

3. Attribute Control During Inference:
    a. You mention that your method "seamlessly supports attribute control during inference." However, it seems that this relies on three external models: pre-trained pose, quality, and FR models. Could you clarify what you mean by "seamless" in this context?
    b. Does relying on these external models introduce any constraints or complexities in practice?

---

> ### Author Response · Authors · 2024-11-18
>
> ## Answers of Weaknesses
> ### Limited Scale of Synthetic Data
> These are the summaries of the number of identities in the prior works. The proposed HSFace400K is already **~4x more** than the second largest dataset in this table. Moreover, based on Figure 5, the actual number of well-separated identities in previous works is **much less than these reported numbers**.
> | Prior works    | the number of identities |
> | -------------- | ------------------------ |
> | IDiff-Face     | 10K                      |
> | DCFace         | 60K                      |
> | ID3            | 10K                      |
> | Arc2Face       | 60K                      |
> | DigiFace       | 110K                     |
> | SynFace        | 10K                      |
> | SFace          | 10.5K                    |
> | IDnet          | 10.5K                    |
> | ExFaceGAN      | 10K                      |
> | SFace2         | 10.5K                    |
> | Langevin-Disco | 30K                      |
> | Ours           | **400K**                     |
>
> We have sampled **1M well-separated identity vectors**. We only use 400K to form a dataset because, as shown in Table 2, using more does not improve accuracy. The proposed algorithm is the first method that allows us to effectively generate this large number of well-separated identities, and having a larger number of identities is recognized as an important factor for FR dataset generation. Thus, this is one of the major contributions rather than a weakness of our work.
>
> ### Lack of Comparison with Larger Datasets
> We do not compare with larger-scale datasets because the highest accuracy we achieved, although higher than prior works, is not higher than CASIA-WebFace. This indicates that many problems still need to be investigated and solved, but generating a large number of well-separated identities is not on the list because this work has solved it. **We will try to add WebFace4M and Glint360K results to the paper.**
>
> ### Unclear Practical Motivation
> The images in the existing training sets are web-scraped from the Internet, which raises severe privacy concerns. Hence, the regulations, such as GDPR, CCPA, PIPL, BIPA, CUBI, have been established to restrict biometric data usage. Given these concerns and similar to previous works, the goal of this work is to generate high-quality FR training sets with identities that do not exist to address the privacy issue. **We will clarify the motivation in the introduction for people who do not know this field well.**
>
> ### Missing Citation
> Thanks for pointing this out and we will add it in our related work.
>
> ## Answers of Questions
> ### Choice of Pre-trained FR Model
> Using a pretrained FR model is a common choice in this field, such as SynFace, DCFace, IDiff-Face, Arc2Face, CemiFace, ID3, etc. Different from these works, we use a FR model is because it can provide well-grouped features, not for evaluating the image reconstruction. The proposed model aims to leverage the potential of high-dimensional space to generate well-separated identities and what it does is convert vectors to images. Hence, it is possible to get rid of the FR model from the pipeline as long as the high-dimensional vectors have well-defined patterns and can map to the images. This is the first work achieving the feature-to-image generation, so investigating how to drop the FR model is not in our scope. Overall, FR model is not a must but a convenient option for this first try.
>
> ### Number of Identities
> We have generated 1M well-separated identity vectors, which can be converted to images. However, we stopped at 400K because HSFace400K does not bring more accuracy on test results than HSFace300K. There are two main reasons to explain this: 1) ViT-Base might not be powerful enough to generate high-quality images, and 2) the intra-class identity consistency is not as high as that of the real dataset. Despite these problems, the proposed datasets achieve **state-of-the-art accuracy**, showcasing the efficacy.
>
> ### Attribute Control During Inference
> There are two main approaches for facial attribute editing: Style transfer and ControlNet. The former cannot achieve explicit attribute control and can lose identity information. The latter needs additional training and external models when a new attribute editing is needed. AttrOP allows us to do plug-and-play attribute operation, with identity preservation and no additional training. In this paper, we control quality and pose by using a pose and quality estimator. This means, any other attributes can be controlled, either separately or combined, in the same way with different attribute models.
>
> The limitation of AttrOP is that when creating an image with extreme attribute conditions, *e.g.*, head pose = 90 degree, it needs more time to search for a proper vector. Fortunately, a dataset does not need a huge number of faces with extreme attributes, so it is not much of an issue.

---

> ### Author Response · Authors · 2024-11-25
>
> Dear Reviewer G33d,
>
> Thanks for your effort in reviewing our work.
>
> We believe there is a misunderstanding of the motivation of this work, which led you to think our work is not worth acceptance.
>
> Yes, there are datasets with real **web-scraped** images available that can be used to train good face recognition models. However, **these web-scraped images raise a severe privacy concern**. Hence, a lot of works try to generate synthetic identities to form a dataset that can be used to train a face recognition model. Our work has the same motivation.
>
> With this motivation, our work has a big contribution to solving the limitation of well-separated identity generation. Also, the proposed datasets achieve state-of-the-art accuracy.
>
> We sincerely hope you can reconsider the contributions of this work and welcome any further questions.

---

> > ### Comment · Reviewer_G33d · 2024-11-26
> > **Reply to author's feedback**
> >
> > I would like to thank the authors for their diligent efforts in addressing all of the reviewers' concerns and conducting additional experiments.
> >
> > The authors have satisfactorily answered all of my questions and concerns. However, I would like to clarify two points:
> >
> > 1. Privacy Leakage: While I understand that generating high-quality synthetic data is a common practice to mitigate privacy risks associated with real-world datasets, the use of WebFace4M and Glint360K, both collected from the internet, raises potential privacy concerns. The authors should explicitly address reviewer oM4G's query regarding the possibility of privacy leakage in the images generated by Vec2Face. Specifically, they should clarify whether any generated images exhibit similar appearances or identities to those in the training datasets, WebFace4M or Glint360K.
> >
> > 2. "Unlimited Number of Well-Separated Synthetic Identities": The claim of an "unlimited number" of well-separated synthetic identities is quite strong and requires further experimental validation. While it is relatively straightforward to sample numerous well-separated feature vectors, the quality of the resulting generated dataset is paramount. The current experiments indicate that increasing the number of identities beyond 400K does not significantly improve FR accuracy, which seems to contradict the initial claim. The authors should consider using a more conservative term like "large" and provide additional evidence to support their assertion.

---

> > > ### Author Response · Authors · 2024-11-27
> > >
> > > Dear Reviewer G33d,
> > >
> > > We are pleased that our answers address your concerns!
> > >
> > > ## Privacy Leakage
> > > We have conducted an experiment regarding identity leakage in Section 4.4, beginning with **'Does Vec2Face generate new identities compared to its training identities?'**. Since this experiment only compares the generated identities with the data used for Vec2Face training, we conducted a further experiment to compare them with all the identities in the WebFace4M dataset.
> > >
> > > Experiment steps:
> > > 1. We use a pre-trained FR model to calculate the identity features for each identity in the datasets listed below.
> > > 2. We use FAISS to search for the highest identity match for each identity vector in the dataset listed below.
> > > 3. We use three similarity threshold values to evaluate the identity leakage by calculating how many identities have a similarity value higher than the threshold when compared to the highest WebFace4M identity match.
> > > 4. We report the percentages.
> > >
> > > | Datasets\threshold |  0.4   |  0.6   |  0.7   |
> > > | :----------------: | :----: | :----: | :----: |
> > > |   CASIA-WebFace    | 21.03% | 7.93%  | 7.87%  |
> > > |      CemiFace      | 95.22% | 59.36% | 24.21% |
> > > |     HSFace10K      | 14.32% |  8.9%  | 8.88%  |
> > > |     HSFace300K     | **1.49%**  |  **0.6%**  |  **0.6%**  |
> > >
> > > This result shows that CemiFace has a severe identity leakage, and our dataset has some identity leakage but can be fixed by dropping them and sampling more identities. For example, we can drop the affected 14.32% of identities and resample an equal number of new identities to replace them.
> > >
> > > For your information, we can generate at least 5M identities which have small identity leakage between WebFace4M and CASIA-WebFace, so that the strategy we mentioned above to avoid the identity leakage is applicable. The detailed explanation of this table can be found under Reviewer oM4G.
> > >
> > > | 5M randomly sampled IDs |  0.4  | 0.6  | 0.7  |
> > > | :---------------------: | :---: | :--: | :--: |
> > > |        WebFace4M        | 1.27% |  0   |  0   |
> > > |      CASIA-WebFace      | 0.17% |  0   |  0   |
> > >
> > > ## "Unlimited Number of Well-Separated Synthetic Identities"
> > > Thanks for your suggestion. We will change "unlimited" to "large".
> > >
> > >  ## Increasing the number of identities to 400K does not further increase the accuracy
> > > This is understandable, because having a large number of well-separated identities is not the only factor to form a good dataset. There are several other factors, including large intra-class variation, high intra-class consistency, etc. In face, we find that all synthetic datasets have lower intra-class identity consistency compared to real datasets, so we believe this is another aspect on which the field should focus. Details are under Reviewer XvoX and this is one of our future works.

---

> > > > ### Comment · Reviewer_G33d · 2024-12-02
> > > >
> > > > Thank the authors for their feedback. I have raised my score from 3 to 5.

---

> > > > > ### Author Response · Authors · 2024-12-02
> > > > >
> > > > > We appreciate your decision!

---

### Official Review · Reviewer_oM4G · 2024-10-31

**Soundness:** 1
**Presentation:** 3
**Contribution:** 2
**Rating:** 5
**Confidence:** 4

**Summary:**

In this paper by using a Masked AutoEncoding on the feature space and a separate FR system, Pose Estimator and Face Quality assessment network authors introduced a pipeline for sampling identities to generate synthetic face dataset for training FR systems.

**Strengths:**

* The paper is well written and the experiments are exhaustive and insightful both in the appendix (especially the analysis of the attributes of the synthetic data) and the main content.
* Applying MAE on the feature space and formulating it to work with the sampling of the random vectors is quite interesting.

**Weaknesses:**

* As the main point of the paper is to generate a synthetic FR dataset, my main concern is that the authors have used the FR model trained on MS1MV2 and Glint360K, and a large subset of the WebFace4m for training/sampling from their method. Training an FR model on  these datasets will probably outperform the FR model trained on the CASIA-WebFace (this dataset used for comparison in Table.1 and Table .2). Later by generating the datasets in a controlled manner they still did not perform as good as a model trained on the CASIA-WebFace, although they are close in most of the benchmarks, so what is the main problem that the authors trying to solve? Why we can not use the CASIA-WebFace, Glint360K, MS1MV2, or the subset of the large WebFace4M that authors used for training their method which probably performs better? Is it trying to address the privacy issues? Are the generated identities different from the ones presented in the dataset used?
Authors may consider the goal of synthetic dataset generation, especially when their approach assumes the existence of a high-quality FR system or a large-scale dataset already available for training their methodologies.
I will consider raising my score if the authors can argue a benefit of this work over the datasets and models they used for training their methodology

* Maybe evaluation using more benchmarks like TinyFace might be needed.
* There is a good amount of studies that show the InceptionV3 trained on the Imagenet is not a good candidate for reporting Frechet Distance:
   [1] The role of Imagenet classes in Frechet inception distance: https://arxiv.org/pdf/2203.06026

   [2] Exposing flaws of generative model evaluation metrics and their unfair treatment of diffusion models: https://proceedings.neurips.cc/paper_files/paper/2023/hash/0bc795afae289ed465a65a3b4b1f4eb7-Abstract-Conference.html

* Some minor Typos:
  [366] surround the pose with braces to appear in the subscript.

  APP[188] five test sets -> Five Synthetic Data methods maybe?

**Questions:**

* There was no reproducibility section in the paper do the authors plan to publish their code/ dataset?
* I was wondering if the authors considered using random perturbed vectors instead of features from an FR model in the training phase.
* As the authors used the subset of WebFace4M (50K identities) for training their method, have authors considered using an FR system trained on this dataset for the ID loss in their proposed method? In this case, one of the weaknesses that I mentioned would be in some extent addressed, if authors can observe the same trend in the results.
* The authors used an FR system trained on MS1MV2,  Glint360K, in their training objective prior, it would be informative if the authors could add the performance of such a system in Table 2.
* In Table.2, I am curious to see what will happen if we add more synthetic images to the CASIA, let’s say CASIA+HSFace200K.
* [392] As the improvement is on the order of 0.1% could this be also the case if authors changed the network, seed, or margin loss?

---

> ### Author Response · Authors · 2024-11-15
>
> We genuinely appreciate your valuable comments and hope our feedback answers the questions properly.
>
> ## Answers of Weaknesses
> ### The purpose of this work
> The images in all the existing large training sets are web-scraped from the internet. This causes a severe privacy issue for those identities in the dataset. Hence, similar to the motivations in the previous works, we aim to generate large-scale face recognition (FR) training sets with identities that do not exist to avoid this privacy issue. Moreover, letting FR models obtain higher accuracy from the generated FR dataset than previous works is another goal. **We will clarify this in the abstract and introduction sections.**
>
> ### Do the proposed HSFace datasets have the identities in the training set of Vec2Face model?
> In the discussion section (line 512-518), we found that there were some training identities at the first stage of identity sampling, but we dropped those training identities when we assembled the datasets. Therefore, all the FR models in this paper were trained using identities that do not exist. Moreover, we want to emphasize that our approach is the first work that can generate this large number of well-separated synthetic identities. **It is a big breakthrough in this field,** because our algorithm solves the limitation of generating large numbers of well-separated identities found in previous works. This allows the following work to compare with larger versions of real FR training sets.
>
> ### InceptionV3 trained on ImageNet is not a good candidate for reporting Freche Distance
> Thanks for pointing this out and we will add these references in our paper. In addition, besides the FID measurement, our dataset is the best at identity separability, identity consistency, and most importantly FR accuracy on test sets. These metrics are really important in synthetic dataset generation.
>
> ### Some minor typos
> Thank you so much! We will fix that typo and double-check the verbal. The "five test sets" in our paper points to LFW, CFP-FP, CALFW, CPLFW, and AgeDB-30, which are the commonly used test sets in FR model performance evaluation. **We will make it clearer in the paper.**
>
> ## Answers of Questions
> *Question1* \
> Yes, we plan to publish the code, model weights, and datasets. The link will be shown in the paper after the publication.
>
> *Question2* \
> It is an interesting research direction! The purpose of using a FR model is that we need the characteristics of the extracted features. Specifically, the image features of the same identity should be clustered together and be away from other identities, so that the Vec2Face model can learn how to map high-dimensional features to their images. With this requirement, using randomly perturbed vectors to train Vec2Face is possible, as long as it is a one-vector-to-one-image mapping. However, it needs extensive experiments to find out how to properly map the perturbations to the intra-class image variations. We love this suggestion and it can be one of our future works.
>
> *Question3*\
> For the ID loss, it was reduced rapidly during training, down to 0.01 within 10 epochs. We have tried to use other face recognition models and the trend is the same, so it have marginal effect if use the other FR models for the ID loss. **We will add it to the Appendix.**
>
> *Question4*\
> We only use the FR model trained with Glint360K, and its performance is shown in Table 1 of section A.7 in the appendix.
>
> *Question5*\
> We are curious about this too and these are the results.
> |      Real + Synthetic      | Average accuracy |
> | :------------------------: | :--------------: |
> |       CASIA-WebFace        |      94.79       |
> | CASIA-WebFace + HSFace10K  |      95.50       |
> | CASIA-WebFace + HSFace20K  |      95.67       |
> | CASIA-WebFace + HSFace100K |      95.69       |
> | CASIA-WebFace + HSFace200K |      95.43       |
> | CASIA-WebFace + HSFace300K |      95.72       |
> | CASIA-WebFace + HSFace400K |      95.64       |
>
> It shows that adding more synthetic data does not bring more accuracy imporvement. It might be because the the domain gap between generated images and real images. We think a stronger generative model might mitigate it.
>
> *Question6:* \
> With a much smaller generative model and a training set that is 1/42 the size of Arc2Face's dataset,, we believe this is not the ceiling of the proposed model. However, our hardware does not support us to do larger scale training. Moreover, one big improvement is that the proposed method allows us to generate unlimited number of **well-separated** identities, so that we can easily scale the dataset up.Generating more images is trivial but not generating more well-separated identities. Eventually, 93.52% is the highest accuracy our work achieves and it is the state-of-the-art with no doubt.

---

> > ### Comment · Reviewer_oM4G · 2024-11-26
> >
> > Thank you for your thorough response.
> >
> > As the authors have highlighted, the primary aim of synthetic face data generation is to address privacy concerns and challenges associated with collecting large datasets. However, in this work, the systems trained on such data are used in the generation pipeline, and samples with high similarity to these datasets are dropped (authors’ first point). Yet, there is no mention of comparisons with these systems or any clear benefits introduced by this approach.
> >
> > Authors need to address: What is the advantage of this synthetic dataset over Glink360 or a subset of WebFace4M? If I understand correctly, there appears to be a significant gap between the WebFace4M (50K identity subset) and Glink360-trained model and the HSFace (A.7—thank you for clarifying this).
> >
> > The main purpose of Arc2Face was personalization and editing, not FR, even though they included FR results in their paper.
> >
> > While the authors claim SOTA results, it’s worth noting that they also used stronger priors and more data, leaving the question open: is the method effective on its own, or are the results largely due to the priors and additional data?
> >
> > Lastly, please revisit my third question for further clarification.

---

> ### Author Response · Authors · 2024-11-25
>
> Dear Reviewer oM4G,
>
> We appreciate your suggestions and hope everything goes well with you.
> Since we both put our effort into the reviewing and rebuttal, we would like to know if our answers address your concerns well and if it is possible to increase the score.
>
> We are here for any additional questions!
>
> Sincerely,
> Authors

---

> ### Author Response · Authors · 2024-11-27
>
> # The goal of synthetic dataset generation in the FR field
> We want to clarify the goal of the field that this paper is related to.
>
> ## **Problem - the web-scraped data raises severe privacy issues.**
> Millions of images are web-scraped online without getting permission from those identities in the dataset, raising privacy concerns in the face recognition field. **This issue is related to all the existing large-scale datasets,** including the Glint360K and WebFace4M you mentioned.
>
> ## **Solution - use synthetic identities (identities that do not exist)**
> The most intuitive way is to generate a dataset that **does not contain any real identities**. This is what our work and related work are trying to achieve. Thus, to answer your question *"What is the advantage of this synthetic dataset over Glink360 or a subset of WebFace4M?"*, **our datasets do not contain any of the real identities, so there is no privacy issue.**
>
> ### Experiment for illustration
> Step 1: We randomly generate 5M vectors that have a similarity value less than 0.3 compared to each other.\
> Step 2: We use Vec2Face (trained with CASIA-WebFace, for fair comparison with other methods) to generate images with the AttrOP to ensure the ID of the generated images aligns with the sampled vectors.\
> Step3: We calculate the cosine similarity between the features of the generated identities and the identity vectors of the real datasets.\
> Step4: We use different similarity threshold values to measure the identity leakage. 1.27% means that 1.27% of the 5M identities have a similarity value higher than 0.4 when compared to the WebFace4M identities. It is small and easy to drop. When we generate the image for each identity, we add small perturbations so that the identity information is not lost. All these ensure the **identities in our dataset do not exist in these real datasets**.
>
> | 5M randomly sampled IDs |  0.4  | 0.6  | 0.7  |
> | :---------------------: | :---: | :--: | :--: |
> |        WebFace4M        | 1.27% |  0   |  0   |
> |      CASIA-WebFace      | 0.17% |  0   |  0   |
>
> ### **Limitations of existing works**
> 1. Cannot achieve the same average accuracy on test sets. However, our work achieves better performance than CASIA-WebFace on several test sets at the same dataset scale, marking the first such achievement in this field. These results are in Table 1 of the main paper and the tables under Reviewer XvoX.
> 2. Cannot generate a large number of well-separated identities. **This limitation is solved by the proposed Vec2Face model.**
> 3. Cannot maintain intra-class consistency while increasing the intra-class variation. This problem is first revealed in our work, so there is still work to follow.
> 4. The approach highly relies on the pre-trained FR model. This is a common limitation of all works in this field, but it is understandable because the field has just begun addressing this problem.
> # Your concerns
> ### The main purpose of Arc2Face was personalization and editing
> We know, but it has state-of-the-art performance, so we need to compare our result with it.
> ### Is the method effective on its own, or are the results largely due to the priors and additional data?
> Here are the FR results for the Vec2Face model trained with CASIA-WebFace, ensuring a fair comparison with other works.
> |    Datasets    | Accuracy  |
> | :--------: | :-------: |
> |   DCFace   |   89.56   |
> |    ID3     |   89.80   |
> | HSFace10K  |   89.67   |
> | HSFace80K | **92.77** |
>
> As we mentioned under Reviewer LGwK, 0.5M data makes the model underfit, so the performance is not as good as the results reported in the paper. First, in 0.5M scale (DCFace, ID3, HSFace10K), our model still has state-of-the-art accuracy because ID3 became available on arXiv after the ICLR deadline. Second, our model allows us to scale the dataset up to increase the FR accuracy, which is not possible in any existing papers. With this short period of time, we only scale the dataset to 4M with 80K identities and the accuracy is significantly higher than ID3. Third, since CemiFace has a severe identity leakage issue, we are not comparing with it. Details can be found under Reviewer LGwK.
>
> ### please revisit my third question for further clarification.
> Within this short period of time, we are not able to retrain another model to address this concern, but **we will do this experiment and include the results in the Appendix.**
>
> ### Benefit of our work vs. existing datasets and models
> 1. Dataset: The proposed datasets do not have real identities, which solves the privacy issue.
> 2. Model: Although the accuracy gap has been reduced in this work, we admit that this gap is still large, and thus there is no advantage from the model performance perspective when comparing with the existing FR systems. However, eliminating this accuracy gap is the final goal of this field and we are still progressing toward it. This work makes a big contribution in the identity generation aspect, accelerating the progress to the final goal.

---

> > ### Comment · Reviewer_oM4G · 2024-12-02
> >
> > Thank you for the detailed response and additional experiments.
> >
> > The significant performance drop when switching to CASIA-WebFace (assuming the reported accuracy is an average over LFW, AgeDB, etc.) suggests that the impact might be even greater if the ID condition were also changed, although the improvements here are not statistically significant.
> >
> > If your point is that the dataset you are creating is devoid of real identities and thats the point of SFR, it’s also important to show that models trained on real data can still expose identities from their training sets to some extent.
> >
> > Based on your experiments, I’ve raised my score but still cannot recommend acceptance for this paper.

---

> > > ### Author Response · Authors · 2024-12-02
> > >
> > > Thank you for giving us your feedback before the deadline.
> > > ## Accuracy decreasing
> > > Here is a detailed comparison between FR models trained with two versions of HSFace10K: 1) the first version was generated by Vec2Face trained with 1M WebFace4M images, and 2) the second version was generated by Vec2Face trained with CASIA-WebFace.
> > >
> > > |     Vec2Face training set      |  LFW  |  CFP-FP   | CPLFW | AgeDB-30 | CALFW | Avg.  |
> > > | :--------------------: | :---: | :-------: | :---: | :------: | :---: | :---: |
> > > |     1M (WebFace4M)     | 98.87 |   88.97   | 85.47 |  93.12   | 93.57 | 92.00 |
> > > |  0.5M (CASIA-WebFace)  | 98.45 |   82.31   | 84.02 |  90.97   | 92.62 | 89.67 |
> > > | Accuracy difference (1M - 0.5M) | 0.42  | **6.66%** | 1.45  |   2.15   | 0.95  | 2.33  |
> > >
> > > This table shows that the accuracy difference in the CFP-FP test set, which contains the **frontal-profile** pairs, is significantly larger than in other test sets. The images with profile head pose in the generated dataset do not adequately represent real profile head poses, causing the trained FR model to perform poorly on the CFP-FP test set. As previously mentioned, this occurs because the Vec2Face model is **underfitting on the profile image reconstruction** due to the limited size of the training set and number of profile images. If CASIA-WebFace contained sufficient images with profile poses, we believe the effect of training size would not be this significant.
> > >
> > > As for the ID condition, the original Vec2Face training set has 1M images from **50K identities**, but CASIA-WebFace has only 0.5M images from **10K identities**, so our further experiments include the effect of the change in the number of IDs.
> > >
> > > Overall, we believe the accuracy decreasing in this experiment is mainly because of the insufficient training images with profile head pose not the number of identities.
> > > ## Identity leakage
> > > Your concern that 'it's also important to show that models trained on real data can still expose identities from their training sets to some extent' points to the identity leakage issue. We conducted further experiments to address this concern.
> > >
> > > For better understanding, let's refer to the identity-cleaning operation in the WebFace4M paper [1] (Section 3, paragraph 4): identity similarity between 0.5 and 0.7 is too vague to determine if they are the same identity, while similarity > 0.7 can be regarded as the same identity.
> > > ### **Random identity generation**
> > > We calculated the similarity between 5M randomly generated identities, 200K WebFace4M identities, and 10K CASIA-WebFace identities.
> > > | 5M randomly sampled IDs |  0.5  | 0.7  |
> > > | :---------------------: | :---: | :--: |
> > > |        WebFace4M        | 0 |  0   |
> > > |      CASIA-WebFace      | 0 |  0   |
> > >
> > > The results show that **none of the randomly sampled identities** has a similarity higher than 0.5 when compared with the real identities. Although there must be identity leakage when we sample even more identities, 5M identities is already larger than the largest dataset with real images, which contains 2M identities. However, since these 5M identities are newly sampled and not directly related to the proposed datasets in the main paper, we conducted a further experiment on the proposed datasets.
> > > ### **Proposed datasets**
> > > We conducted the same experiment between the identities in the proposed HSFaces and WebFace4M identities.
> > > | Datasets\threshold |  0.5   |  0.7   |
> > > | :----------------: | :----: | :----: |
> > > |     HSFace10K      | 9.01% | 8.88%  |
> > > |     HSFace300K     | 0.61%  | 0.6%  |
> > >
> > > This result shows that identity leakage exists in the proposed dataset. However, it can be easily solved by replacing these identities with newly sampled ones. After replacing these 9.01% identities in HSFace10K, there is no identity overlap and we obtain **91.93%** FR accuracy. This indicates that while identity leakage can exist in the generated dataset, our method allows us to drop these detected identities without reducing accuracy, **which was impossible in previous work!**
> > > ### **Using the FR model trained with 1M WebFace4M to calculate the ID loss during Vec2Face training**
> > > We are running the training experiment with the FR model trained on 1M WebFace4M images, but we will not be able to provide the results before the deadline. Based on the ID loss values obtained so far, there is no difference from the previous training, so we believe the accuracy trend will be the same.
> > > ### **The position of our work in SFR**
> > > This is a new but important field, as evidenced by recent works accepted at ICCV2023 (DCFace, IDiff-Face), ECCV2024 (Arc2Face), and NeurIPS2024 (CemiFace, ID^3). We admit that our work does not solve all the problems, but we want to emphasize that our work is the best by far with respect to FR accuracy, dataset scalability, and identity leakage.
> > >
> > > We hope we have addressed all your concerns properly this time.
> > >
> > > Reference: \
> > > [1] WebFace260M: A Benchmark Unveiling the Power of Million-Scale Deep Face Recognition

---

### Official Review · Reviewer_XvoX · 2024-11-02

**Soundness:** 3
**Presentation:** 3
**Contribution:** 4
**Rating:** 8
**Confidence:** 4

**Summary:**

The paper introduces a novel Vec2Face generative framework for generating identity-specific synthetic face images suitable for training deep recognition models. To this end, the authors rely on a feature Masked Auto-Encoder (fMAE) architecture, which is trained to produce face images of a desired identity based on input features extracted from a pretrained Face Recognition (FR) model. However, these identity features are first projected and expanded to 2D feature maps to suit the nature of the fMAE architecture. Training is performed with four objectives, including the standard reconstruction and perceptual loss functions, along with an identity-based loss, which enables better identity consistency, and a GAN loss that utilizes a patch-based discriminator to ensure sharper images. During inference, the approach utilizes PCA for sampling unique identity vectors, upon which perturbation is performed to obtain diverse samples of a specific identity. Additionally, the approach also supports control over the pose and quality, through guided vector perturbation. The authors showcase the suitability of their approach through extensive experiments across multiple datasets, with a focus on training deep face recognition models with the generated synthetic data. The proposed approach outperforms existing state-of-the-art methods in terms of the accuracy achieved with a trained recognition models on standard verification benchmarks. Importantly, Vec2Face also enables better scalability than existing approaches, thus allowing the generation of diverse datasets of up to 20 million images, compared to existing datasets of 500 thousand images. Furthermore, the authors also perform sufficient ablation studies for supporting the various choices of their particular solution.

**Strengths:**

The submitted paper is writte well and presents the problem and proposed solution in a concise and clear manner. The proposed approach improves on the state-of-the-art with a unique fMAE architecture, differentiating itself from existing diffusion and GAN-based solutions. Differently, it also proposes to utilize guided perturbation during inference, to not only achieve higher intra-identity diversity of samples but also allow for control over the pose and quality of images, which is crucial for generating large-scale datasets suitable for training recognition models. As a result, the recognition model trained on the generated data of Arc2Face even achieves better results on the Cross-Age LFW benchmark than when training with the real CASIA-WebFace dataset. This represents a tremendous achievement as existing approaches typically achieve worse performance. The proposed approach also enables scaling the generated datasets without drastically impacting inter-class separability, in turn achieving significantly larger dataset sizes than existing approaches. Overall, the paper is suitable for the conference and will likely highly influence future research on the generation of synthetic biometric datasets.

**Weaknesses:**

Despite the many strengths of the paper, there are a few areas where the paper could be further improver. The evaluation of the approach mainly revolves around the accuracies achieved with trained recognition models on existing benchmarks (apart from Table 4 and Figure 5). While this focus is understandable, it would be beneficial to also analyse other aspects of the data.

1.	Face Image Quality Assessment (FIQA) measures could provide additional insight as they are specifically designed for evaluating the quality of face images and their suitability for recognition tasks. Examples of recent FIQA measures with publicly available implementations include CR-FIQA (Boutros et al., CVPR 2023) and DifFIQA (Babnik et al., IJCB 2023).

2.	Comparison with the state-of-the-art (e.g. Table 4) could also be expanded with metrics that measure fidelity and diversity separately could also be utilized to provide a deeper understanding of the generated datasets, e.g., improved precision and recall (Kynkäänniemi et al., NeurIPS 2019).

3.	In the supplementary material when comparing genuine and imposter distributions it would also be beneficial to provide results obtained with a real-world dataset to provide the baseline of desired distributions suitable for training face recognition models.

4.	The utilized verification benchmarks are of a rather small scale (roughly 3000 genuine and imposter pairs). It might be beneficial to evaluate the approach on a more complex and larger-scale set, e.g. the IJB-C verification benchmark. There the difference performance improvements of the Vec2Face method could be further highlighted.

Lastly, Despite the detailed nature of the paper, the utilized patch-based discriminator is rather poorly explained. More information should be added to the paper or at least to the supplementary material. For example, what sort of architecture does it utilize? Is it initialized with a pretrained model? The formulation of perceptual and GAN loss functions could also be provided.

**Questions:**

1.	Despite the statement provided in Section 3.4., the novelty introduced with this paper remains slightly unclear. The overall paper gives the impression that the fMAE is novel, however, Section 3.4. does not seem to point this out. Can you perhaps elaborate more on the novelty of the fMAE? Does it share similarities or draw inspirations from previous works, namely [1]? Is the novelty perhaps the projection and expansion of input features to 2D feature maps?

[1] Hu, Junhao, et al. "Features Masked Auto-Encoder-Based Anomaly Detection in Process Industry." 2023 IEEE 12th Data Driven Control and Learning Systems Conference (DDCLS). IEEE, 2023.

2.	When discussing the influence of inter-class separability on the accuracy of trained recognition models, it is observed that a too large separation does not benefit the performance. However, this might not be a unique finding, as results in [2] also showcase that well-separated synthetic identities actually result in lower verification performance. Can you comment more on this phenomenon? Does this mean that future research will have to determine the best balance between inter-class separability and intra-class variation?

[2] Boutros, Fadi, et al. "IDiff-Face: Synthetic-based face recognition through fizzy identity-conditioned diffusion model." Proceedings of the IEEE/CVF International Conference on Computer Vision. 2023.

---

> ### Author Response · Authors · 2024-11-18
>
> We sincerely thank you for recognizing our contribution to this field!
> ## Answers of Weaknesses
> ### Using different quality evaluators
> Using other FIQA measures in AttrOP is definitely worth trying! We will extend this work to a longer journal and **this suggestion will be included in future work.**
>
> ### Precision and recall
> These are the values of precision and recall for reconstructing LFW and Hadrian images. These two metrics might be good for evaluating the object generation results (*e.g.*, ImageNet), but face recognition is more class/ID-oriented rather than pixel-oriented. We think evaluating the model by cosine similarity, as shown in Figure 1 of the Appendix, is a better indicator for the FR model training task.
>
> | precision/recall |  Hadrian  |    LFW    |
> | :--------------: | :-------: | :-------: |
> |     Arc2Face     | 0.03/**0.51** | 0.17/**0.46** |
> |     Vec2Face     | **0.05**/0.00 | **0.7**/0.03  |
>
> ### Providing genuine and impostor distribution in Figure 4 of the supplementary material
> We appreciate your suggestion! **We will update Figure 4.**
>
> ### Evaluating the approach on a more complex and larger-scale set
> These are the accuracy results (TPR@FPR=1e-4) on IJBB and IJBC. Surprisingly, we find that the model trained with HSFace10K outperforms the model trained with CASIA-WebFace.
>
> | SE-R50 (0.5M images) | IJBB  | IJBC  |
> | :------------------: | :---: | :---: |
> |    CASIA-WebFace     | 10.77 | 11.86 |
> |      HSFace10K       | **83.82** | **86.96** |
> |        DCFace        | 66.47 | 69.92 |
>
> To ensure the correctness of the code, we run the same measurement on a 5M scale. The accuracy value of the WebFace4M is consistent with that reported in the original paper.
>
> | SE-R50 (5M images) | IJBB  | IJBC  |
> | :----------------: | :---: | :---: |
> |     WebFace4M      | **95.07** | **96.63** |
> |     HSFace100K     | 86.16 | 89.73 |
>
> ### Details of patch-based discriminator
> We refer to the code at this link to form the discriminator:
> https://github.com/lucidrains/parti-pytorch/blob/main/parti_pytorch/vit_vqgan.py#L171. It is trained from scratch. **We will provide a more detailed description of it in the Appendix.**
>
> ## Answers of Questions
> ### Question 1
> We think the biggest novelty of this work is demonstrating feature-to-image generation rather than image-to-image generation for the first time. The comparison between Vec2Face and Arc2Face results shows that feature-to-image generation is more appropriate for synthetic FR dataset generation. Moreover, this allows us to generate an unlimited number of well-separated identities and control the intra-class similarity and variation by controlling the sampled vectors. As for the fMAE, it is a novel design but can be replaced by other advanced architectures. The key differences between MAE, FMAE[1], and ours are that 1) the input is a 2-D feature map, achieved by feature projection and expansion, and 2) we use replace/fill the masked positions by image feature rather than mask tokens, so the model learns image generation purely from the image features. This guarantees the characteristics of the feature, *e.g.*, the ID/class information, can be transferred to the image. **We will elaborate on it in the Appendix.**
>
> ### Question 2
> First of all, both inter-class separability and intra-class variation are important, because improving both factors increases accuracy when they are not already large. We are not sure if we need to determine the best balance between inter-class separability and intra-class variation, but we observe that the datasets with high separability have lower intra-class similarity than the real dataset, *e.g.*, the mean value of intra-class similarity of DCFace is 0.52, HSFace10K is 0.64, and CASIA-WebFace is 0.76. This suggests that the next research direction should focus on maintaining large intra-class variation and inter-class separability while increasing intra-class similarity. **We will add this conclusion to the main paper.**

---

> > ### Comment · Reviewer_XvoX · 2024-11-25
> > **Reply to provided answers**
> >
> > The authors have adequately addressed my raised concerns and provided clarification where necessary. I would also like to thank the authors for running the additional experiments and providing valuable feedback regarding inter-class separability and intra-class variation.

---

> > > ### Author Response · Authors · 2024-11-25
> > >
> > > We really appreciate your valuable comments and suggestions. Your suggestions make our paper stronger.

---

### Meta-Review · Area_Chair_r36v · 2024-12-21

**Metareview:**

The rebuttal provided clarifications about the proposed method and its analysis that were useful for assessing the paper's contribution and responded adequately to most reviewer concerns. After discussion, reviewer XvoX recommended acceptance, LGwK recommended marginal acceptance, G33d and oM4G recommended marginally below acceptance. The AC agreed this work is valuable to the ICLR community. The final version should include all reviewer comments, suggestions, and additional clarifications from the rebuttal.

**Additional Comments On Reviewer Discussion:**

NA

---

### Decision · Program_Chairs · 2025-01-22

Accept (Poster)